



# Instant and delayed effects of march biomass burning aerosols over the Indochina Peninsula

Anbao Zhu[1,2], Haiming Xu[1,2], Jiechun Deng[1,2], Jing Ma[1,2], Shaofeng Hua[3]

[1] Key Laboratory of Meteorological Disaster/KLME/ILCEC/CIC-FEMD, Nanjing University of Information Science & Technology, Nanjing 210044, China
[2] School of Atmospheric Sciences, Nanjing University of Information Science & Technology, Nanjing 210044, China
[3] CMA Weather Modification Centre (WMC), Beijing 100081, China

*Correspondence to*: Haiming Xu (hxu@nuist.edu.cn)

**Abstract.** Through analyzing observations and simulations from the Weather Research and Forecasting model coupled with Chemistry, we investigated instant and delayed responses of large-scale atmospheric circulations and precipitation to biomass burning (BB) aerosols over the Indochina Peninsula (ICP) in the peak emission of March. The results show that the BB aerosols inhibit precipitation over the ICP in March, and promote precipitation from early-April to mid-April. Specifically, the March BB aerosols over the ICP can induce mid-to-lower tropospheric heating and planetary boundary layer cooling, to enhance local atmospheric stability; meanwhile, the perturbation heating can trigger an anomalous low in the lower troposphere to moisten the mid troposphere. However, the convection suppression due to the stabilized atmosphere dominates over the favorable water-vapor condition induced by large-scale circulation responses, leading to an overall reduced precipitation over the ICP in March. For the delayed effect, the anomalous low can provide more water vapor as the monsoon advances in early-April, although it becomes much weaker without BB aerosols' strong heating. On the other hand, the convective instability above 850 hPa is enhanced by more water vapor, resulting in enhanced precipitation over the ICP, northern South China Sea, and southern China. Thereafter, the condensational latent heating gradually takes over from the BB aerosol radiative heating, acting as the main driver for maintaining the anomalous circulation and thus the delayed effect in mid-April.

## 1 Introduction

Biomass burning (BB), including agro-residue burning and forest or prairie fires, is one of the largest sources of many trace gases and aerosol particles in the atmosphere (Reid et al., 2005). Globally, BB contributes 42% of the black carbon (BC) emissions and 74% of the organic carbon (OC) emissions (Bond et al., 2004). Smoke aerosols produced by BB can reduce air quality, diminish visibility and harm public health (Huang et al., 2013; Yadav et al., 2017; Requia et al., 2021). BB-emitted aerosols also have vital impacts on regional climate and hydrological cycle through interactions with radiation, clouds and precipitation (Koren et al., 2004; Jacobson, 2014; Hodnebrog et al., 2016; Liu et al., 2020). The Indochina Peninsula (ICP) is one of the most active fire hotspots in the world (Lin et al., 2009; Gautam et al., 2013; Yadav et al., 2017),


with high population density, thus high social and economic relevance, and with strong monsoon circulation variability (Li et
al., 2016; Wu et al., 2016). Therefore, it is essential to investigate the feedback mechanisms of BB aerosols-climate
interactions to better understand aerosols' climatic and socio-economic impacts (Lau, 2016; Ding et al., 2021).
BB aerosols can affect the climate in several ways. The aerosols, such as BC and OC aerosols, can directly scatter and
absorb solar radiation (i.e., the so-called "direct effect"), thereby reducing the solar radiation reaching the surface. Both
observational and numerical studies suggested that BB aerosols' direct effect can inhibit vertical instability by heating the
atmosphere of the smoke aerosol layer and cooling the surface, thereby reducing surface fluxes and suppressing warm-cloud
formation and convective activity (Koren et al., 2004; Feingold et al., 2005; Hodnebrog et al., 2016; Huang et al., 2016b),
and enhancing low-cloud fraction (Sakaeda et al., 2011; Lu et al., 2018; Ding et al., 2021). On the other hand, BB aerosols
can locally reduce precipitation by serving as cloud condensation nuclei and ice nuclei, increasing cloud droplet number
concentration, decreasing droplet effective radii (i.e., "indirect effect"), and decelerating the autoconversion process (Lee et
al., 2014; Liu et al., 2020; Herbert et al., 2021). Numerical modelling studies have found that the direct effect dominates at
low BB aerosol loading, while the indirect effect dominates at high BB aerosol loading (Liu et al., 2020; Herbert et al., 2021).
However, the initial suppressive effect of BB aerosols on rainfall can lead to convective invigoration by cold rain processes
(Martins et al., 2009). BB aerosols may also enhance rainfall under certain conditions, which are highly dependent on factors
such as the altitude and longevity of the smoke plume (Tummon et al., 2010; Ban-Weiss et al., 2012; Herbert et al., 2021),
the atmospheric degree of instability (Gonçalves et al., 2015) and the diurnal cycle of the convective system (Lee and Wang,
2020; Herbert et al., 2021). The above-mentioned perturbations caused by BB aerosols can also affect large-scale
atmospheric circulation, thus changing the regional climate (Zhang et al., 2009; Lee et al., 2014; Jiang et al., 2020; Zhou et
al., 2021).
The ICP experiences substantial agro-residue burning across farmlands in preparation for planting during the dry season,
typically between February and April with a maximum occurrence in March (Huang et al., 2013; Shi et al., 2014) (Figs. 1a–
e). Large amounts of BB aerosols are injected into the atmosphere, uplifted up to 3-km height by the India-Burma trough and
transported to southern China and the South China Sea (SCS), and even to the western North Pacific Ocean by the
subtropical southwesterly jet (Lin et al., 2009; Huang et al., 2013; Huang et al., 2016a; Zhu et al., 2021). The BB aerosols
become minimal after the monsoon rainfall onset in late-April due to rainout and washout processes (Huang et al., 2016a).
The effects of BB aerosols over the ICP on regional air quality (Lin et al., 2009; Huang et al., 2013; Lin et al., 2014; Yang et
al., 2022a) and climate (Lee and Kim, 2010; Lee et al., 2014; Pani et al., 2018; Dong et al., 2019; Wang et al., 2021; Yang et
al., 2022b) have been widely investigated based on observations and numerical modeling studies. However, aerosol-cloud-
precipitation interactions over the ICP have rarely been explored. Using an atmospheric global climate model (AGCM)
coupled with an aerosol module, Lee and Kim (2010) showed that BC's radiative forcing (including anthropogenic and BB-
emitted) in East Asia induces an anomalous meridional circulation through radiation effect during spring. The anomalous
upward motion near 30 °N causes increased precipitation over Myanmar and Bangladesh, while the anomalous downward
motion around 10 °N causes a decrease in precipitation over Southeast Asia. Based on the Goddard Earth Observing System





version 5 (GEOS-5)/AGCM model, Lee et al. (2014) suggested that both the direct effect (increasing lower-atmospheric
stability) and indirect effect (decelerating cloud droplet autoconversion process) of BB aerosols can suppress local
precipitation in the ICP during the pre-monsoon season (March–April), and the large-scale advection of cloud moisture
invigorates the downwind rainfall. Yang et al. (2022b) utilized the Weather Research and Forecasting model coupled with
Chemistry (WRF-Chem) to show that the increased atmospheric stability induced by BB aerosols inhibits local rainfall over
the ICP. The low-level cyclonic anomaly wind induced by the BB aerosol heating can modify moisture transport, leading to
increased (decreased) rainfall over the southern coast (northern inland) of southern China. A case study by Wang et al. (2021)
revealed that BB aerosols transported from the ICP can suppress convective precipitation and enhance non-convective
precipitation over southern China. Most of these studies focused on the seasonal time scale (Lee and Kim, 2010; Lee et al.,
2014; Yang et al., 2022b) or individual cases lasting a few days (Wang et al., 2021). However, the BB emission over the ICP
has a strong intra-seasonal variability peaking in March (Fig. 1e), whose instant and delayed effects on the climate remain
unclear.

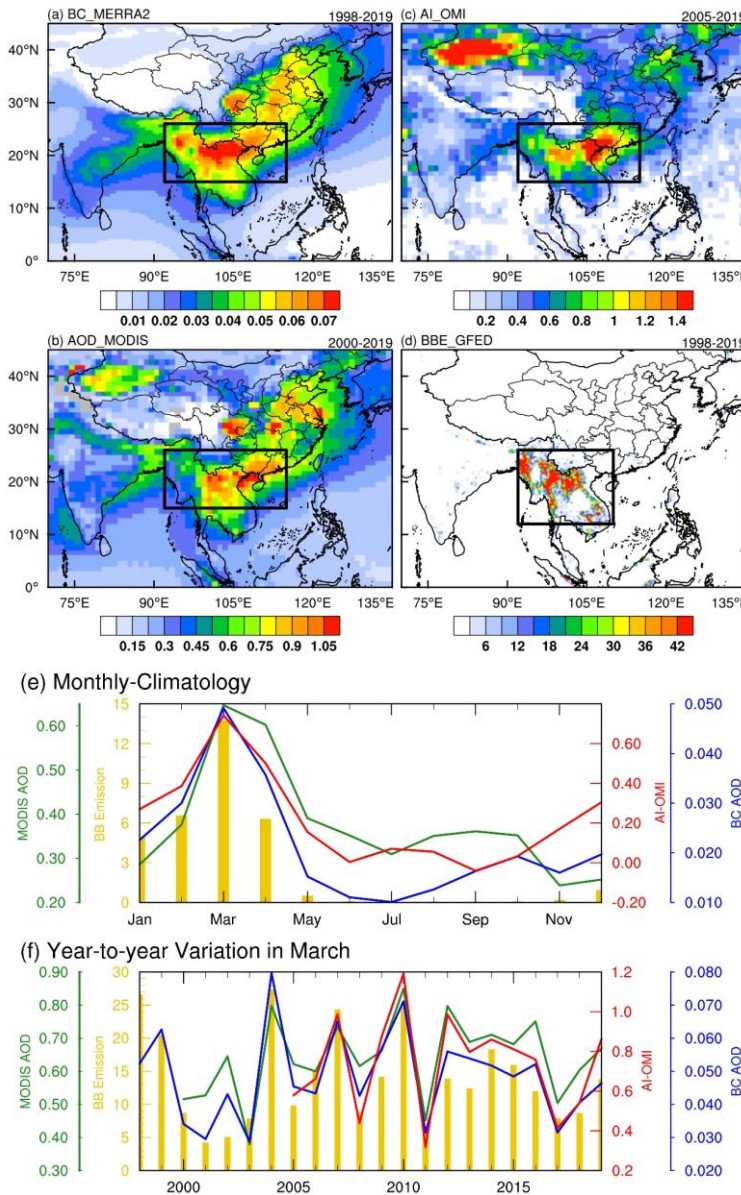

**Figure 1: Spatial distribution of March (a) black carbon (BC) aerosol optical depth (AOD; shading, unitless) averaged over 1998–2019 from MERRA-2, (b) AOD (unitless) averaged over 2000–2019 from MODIS Terra, (c) aerosol index (AI; unitless) averaged over 2005–2019 from OMI, and (d) biomass burning (BB) carbon emission (shading; g C m$^{-2}$ month$^{-1}$) averaged over 1998–2019 from GFEDv4.1. (e) Monthly climatology of BB aerosol indices (blue line for BC AOD, green line for AOD, red line for AI) and emission (gold bar) averaged over Indochina [92°–115°E, 15°–26°N for BC AOD, AOD and AI; 92°–110°E, 12°–26°N for BB emission; as outlined by the black boxes in (a–d)]. (f) Same as (e), but for the time series of monthly averaged BB aerosol indices in March.**

In this study, we examine the impacts of March BB aerosols over the ICP using both observations and model experiments. In particular, we address the following questions: (1) What are the instant and delayed effects of March BB aerosols over the ICP on atmospheric circulation and precipitation? (2) What are the differences between these two effects and what are their





underlying physical mechanisms? The remaining paper is organized as follows. In Sect. 2, we describe the data, methods,
model, and experimental design. In Sect. 3, we present the observed evidence of BB aerosol impacts on circulation and
precipitation. In Sect. 4, we discuss the responsible physical mechanisms based on simulation results. Conclusions and
discussion are provided in Sect. 5.

## 2 Methodology

### 2.1 Data and statistical methods

The meteorological and BC aerosol data used in this study are the Modern Era Retrospective analysis for Research and
Applications Version 2 (MERRA-2) from the National Aeronautics and Space Administration (NASA) Global Modeling and
Assimilation Office (GMAO) (Gelaro et al., 2017), with a spatial resolution of 0.5 °by 0.65 °(longitude by latitude) on 72
levels. MERRA-2 reanalysis is the first satellite era (1980 onward) reanalysis data jointly assimilating meteorological and
aerosol observations. The MERRA-2 aerosol data is produced using the Goddard Chemistry Aerosol Radiation and
Transport (GOCART) aerosol model coupled to the GEOS-5 data assimilation system. The GOCART model simulates five
aerosol species: dust, black carbon, organic carbon, sulfate and sea salt. The GEOS-5 assimilates the bias-corrected aerosol
optical depth (AOD) from the Advanced Very High Resolution Radiometer (AVHRR) instrument over the ocean (Heidinger
et al., 2014), the Moderate resolution Imaging Spectroradiometer (MODIS) from the Terra and Aqua satellites (Levy et al.,
2010), Multiangle Imaging SpectroRadiometer (MISR) AOD over land (Kahn et al., 2005), and ground-based Aerosol
Robotic Network (AERONET) AOD (Holben et al., 1998). Numerous evaluations on the MERRA-2 aerosol data have
shown that both the AOD and the vertical structure of aerosol properties in the MERRA-2 have good agreement with the
observations (Buchard et al., 2017). In this study, we use the monthly mean BC AOD.
We also use the AOD from 1 °MODIS Terra Level-3 monthly product (MOD08_M3) (Gupta et al., 2016), aerosol index (AI)
from 1 ° Ozone Monitoring Instrument (OMI)/Aura Level-3 daily product (OMAERUVd) (Torres et al., 2007) and BB
emissions from the Global Fire Emissions Database version 4.1 (GFEDv4) (Randerson et al., 2017) to compare with
MERRA-2 BC AOD. In addition, we use the atmospheric fields from the fifth generation European Centre for Medium-
Range Weather Forecasts (ECMWF) reanalysis data (ERA5) (Hersbach and Dee, 2016), including zonal and meridional
wind components on 0.25 ° grid. The monthly and daily precipitation data on 0.25 ° grid is from the Tropical Rainfall
Measuring Mission (TRMM) Multi-satellite Precipitation Analysis (TMPA) 3B43 and 3B42 (Huffman et al., 2007),
respectively.
For consistency, the precipitation data from the TRMM, the ERA5 reanalysis data, the GFEDv4 BB emissions, and
MERRA-2 BC AOD all cover the same period of 1998–2019. MODIS AOD and OMI AI cover the periods of 2000–2019
and 2005–2019, respectively. In this study, we focus on the effect of March BB aerosols on regional climate in early-spring
(March 1st–April 20th), including the instant effect in March and the delayed effect in early-April (1st–10th) and mid-April
(11th–20th). The linear- regression analysis is used and subjected to the two-tailed Student's $t$-test for statistical significance.


## 2.1 Model and experimental design

In this study, the WRF-Chem version 4.2.1 is used to simulate the evolution of BB aerosols and trace gases, to investigate their interactions with meteorological conditions over the ICP and East Asia. The model is configured to cover the Bay of Bengal, ICP and East Asia (Fig. 2) with 331 × 255 grids at 27-km horizontal resolution and 42 levels from the ground to 50 hPa. The planetary boundary layer (PBL) processes are parameterized using the Mellor-Yamada-Janjic (MYJ) scheme with local vertical mixing (Janjić, 1994), combined with the Noah Land Surface Model and the Monin-Obukhov scheme for the surface layer physical processes and the interaction with land surface (Chen et al., 2010; Pahlow et al., 2001). The Rapid Radiative Transfer Model for General circulation models (RRTMG) coupled with aerosol radiative effect is used for both shortwave (SW) and longwave (LW) radiation (Iacono et al., 2008). The double-moment Morrison microphysics scheme (Morrison et al., 2009) and Grell-Freitas (GF) cumulus scheme (Grell and Freitas, 2014) are used to ensure that aerosol indirect effects are included. The Carbon-Bond Mechanism version Z (CBMZ) gas-phase chemistry mechanism combined with the Model for Simulating Aerosol Interactions and Chemistry (MOSAIC) aerosol module (Zaveri and Peters, 1999; Zaveri et al., 2008) are selected for aerosol simulation. Aerosol optical properties are calculated based on the Maxwell approximation (Bohren and Huffman, 1983).

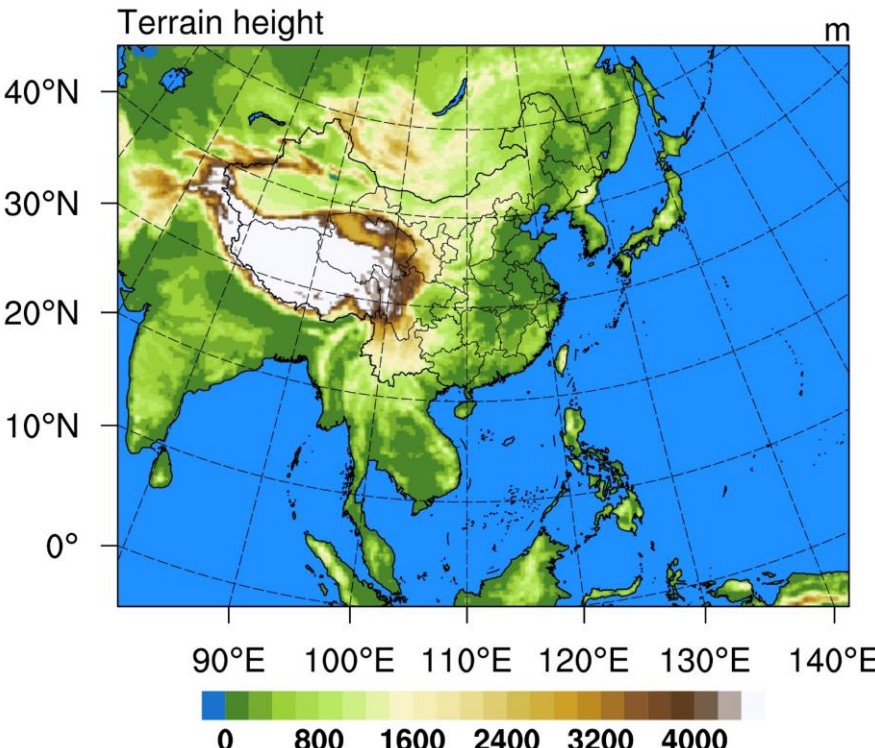

**Figure 2: Model domain and orography (shading; m).**

The boundary and initial conditions of meteorological fields are derived from the National Centers for Environmental Prediction (NCEP) Final Analysis (FNL) data with 1 ° spatial resolution and 6-h temporal interval. The input sea-surface





temperature (SST) data is the NCEP real time global SST analysis. The anthropogenic emission source comes from the
Multi-resolution Emission Inventory for China (MEIC) database for China (Li et al., 2017a) and from the MIX inventory (Li
et al., 2017b) for regions outside of China. The biogenic emissions are calculated online using the Model of Emissions of
Gases and Aerosols from Nature (MEGAN) (Guenther et al., 2012). The GOCART dust emission scheme with the Air Force
Weather Agency (AFWA) modifications (LeGrand et al., 2019) is used to simulate dust emissions. The high-resolution fire
emissions based on the Fire INventory from NCAR (FINN) version 1.5 (Wiedinmyer et al., 2011) are selected as the BB
emissions. Specific settings are listed in Table 1.
**Table 1. WRF-Chem model parameterization option settings and emissions used in this study**

| Option name | Scheme |
| --- | --- |
| Longwave radiation | RRTMG |
| Shortwave radiation | RRTMG |
| Microphysics | Morrison 2-mom |
| Boundary layer | MYJ |
| Cumulus | Grell-Freitas |
| Land surface | Unified Noah |
| Surface layer | MM5 Monin-Obukhov |
| Aerosol chemistry | MOSAIC |
| Gas chemistry | CBMZ |
| Photolysis | Fast-J |
| Aerosol mixing rule | Maxwell–Garnett approximation |
| Dust emissions | GOCART-AFWA |
| Biogenic emissions | MEGAN version 2 |
| Anthropogenic emissions | MEIC for China and MIX for outside of China |
| Biomass burning emissions | FINN version 1.5 |

To investigate the impacts of March BB aerosols on radiation, circulation and precipitation, we conduct two groups of
simulations with different BB emission scenarios and compare these results. The control experiment (CTRL) has the original
BB emissions, while the sensitivity experiment (BBER) has the March BB emissions reduced to 15%. To increase the
robustness of our findings, we use six ensemble members for each experiment by perturbing initial and boundary conditions,
that is, the ensemble simulations start at one day apart on February 20th–25th, 2010, respectively, and all end on April 30th,
2010. Thus, different starting day in February for each member is discarded as spin-up time, and we only focus on the period
from March 1st to April 20th, 2010. We chose the year of 2010 for modeling because the BB emission in 2010 was above the





average and was about six times higher than that in 2001 (the lowest year during 1998-2019 and similar to the BB reduction
used in BBER; Fig. 1f), which is suitable for investigating the effects of BB aerosols on atmospheric circulation and
precipitation.
**3 Observations**
**3.1 Variation in BB aerosols**
For observational evidence of possible responses of atmospheric circulation and precipitation to BB aerosols, we first
examine the spatial distribution of the climatological mean BB aerosols in March (Figs. 1a–d) and their temporal variation
(Fig. 1f) via multiple data sources. The spatial pattern of BB aerosols from the aerosol reanalysis data (MERRA-2) is quite
consistent with multiple satellite retrievals (Figs. 1a–d). The high BC aerosol loading is concentrated in the northern ICP
with a maximum BC AOD exceeding 0.07 (Fig. 1a), which is contributed by BB emissions (Fig. 1d). High BC AOD also
appears over the Sichuan Basin and central-eastern China, likely caused by anthropogenic activities (Qin and Xie, 2012;
Ning et al., 2018). High MODIS AOD values are also seen over northwestern China (Fig. 1b), as large dust aerosols are
emitted from the Taklimakan Desert in March (Bao et al., 2009). As positive AI generally represents absorbing aerosols
(dust and smoke), high AI is found over the northern ICP and northwestern China (Fig. 1c). Unlike the high BC loading over
the Sichuan Basin and central-eastern China (Fig. 1a), the AI is small over these regions likely because the AI's sensitivity to
aerosol amount increase more or less proportionally with the aerosol layer height, while any aerosol below about 1000 m is
unlikely to be detected (de Graaf et al., 2005). The dust and BB aerosols are transported eastward at higher atmospheric
levels and are more easily detected, whereas anthropogenic pollution transport mainly occurs within the boundary layer,
giving rise to smaller AI (Kaskaoutis et al., 2010).
For temporal variation, the BC AOD from the MERRA-2 over the ICP agrees well with satellite datasets and BB emissions.
Figure 1f shows the time series of area-averaged monthly BB aerosol indices in March for the northern ICP (92°–115°E, 15°–
26°N for BC AOD, AOD and AI; 92°–110°E, 12°–26°N for BB emissions). The correlations between the time series of
MERRA-2 BC AOD and MODIS AOD (2000–2019), AI (2005–2019), and BB emission (1998–2019) are 0.90, 0.93 and
0.85, respectively; all are statistically significant at the 99 % level. This indicates that the BB aerosols over the ICP have
large interannual fluctuation in March, consistent with the recent study by Ding et al. (2021) based on multiple satellite
records. However, such interannual variation could be influenced by meteorological factors such as the India-Burma trough
(Huang et al., 2016a) and El Niño-Southern Oscillation (ENSO) (Zhu et al., 2021). On the other hand, the interannual
fluctuation can be used to detect climate effects of the aerosols. Given this, we define a BB aerosol index (BBAI) the time
series of MERRA-2 BC AOD (1998–2019, blue line in Fig. 1f) to explore BB aerosols' effects on atmospheric circulation
and precipitation.





## 3.2 Relationship between BB aerosols and precipitation

Figure 3 shows the regressed anomalies of BC AOD, precipitation and 850-hPa wind upon the BBAI in March and in early-to-mid April. In March, significant positive BC AOD anomalies are seen over the ICP, northern SCS, southern China, and the ocean south of Japan (Fig. 3a), as the BB aerosols emitted from the central and northern ICP are transported eastward by the prevailing winds (Lin et al., 2009; Huang et al., 2013; Huang et al., 2016a; Huang et al., 2020). Correspondingly, the rainfall over the ICP is reduced by anomalous westerly wind, while the rainfall in coastal Southeast China is enhanced by anomalous southerly wind (Fig. 3d), forming a dipole anomaly structure.

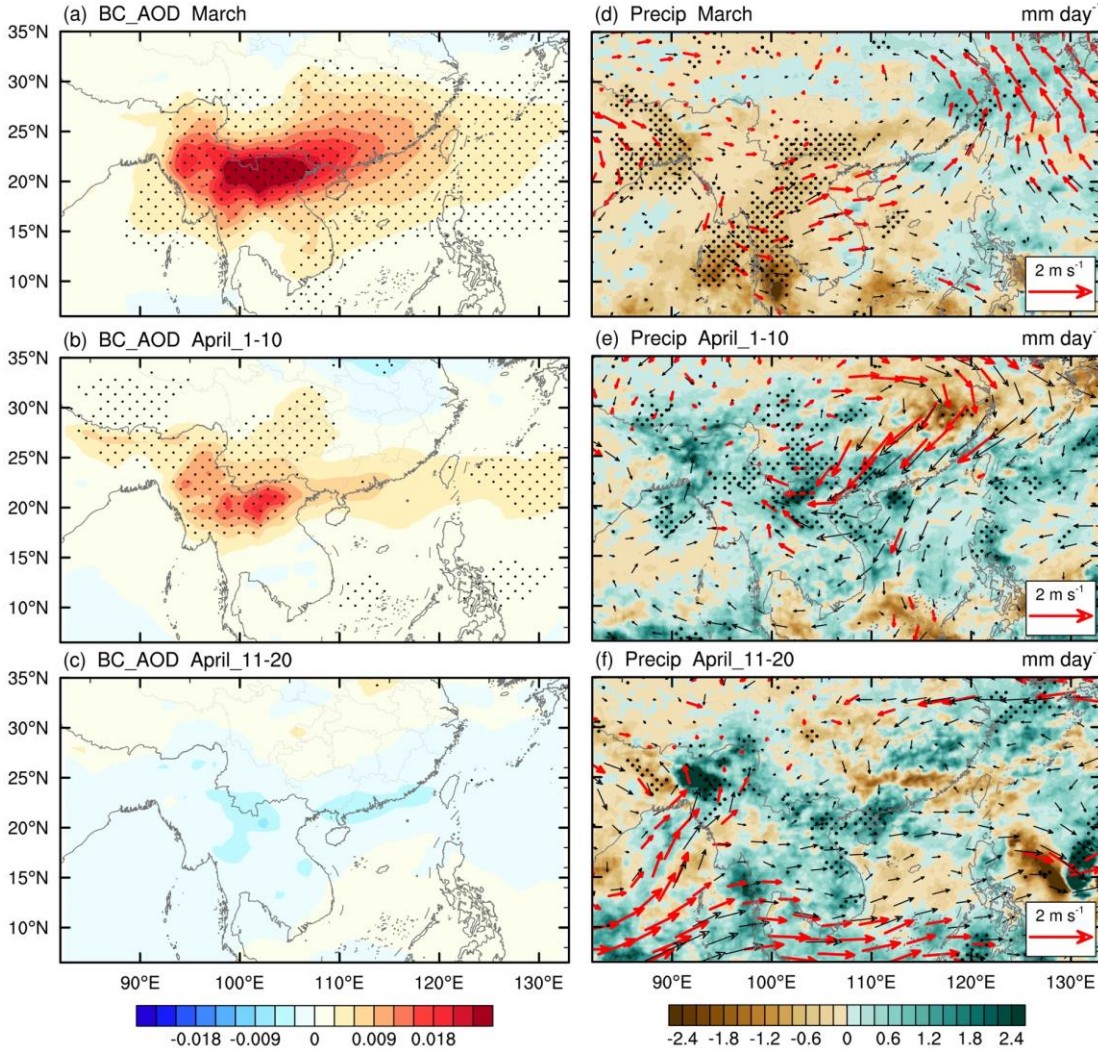

**Figure 3: Regressions of anomalies in (a–c) BC AOD (shading; unitless) and in (d–f) precipitation (shading; mm day⁻¹) and 850-hPa wind (vector; m s⁻¹) onto standardized BBAI in (a, d) March, and in (b, e) early-April and (c, f) mid-April. Stippling (red vector) denotes the regressed anomalies of BC AOD and precipitation (of wind) are statistically significant at the 95% confidence level based on Student's *t*-test.**





Generally, the lifetime of BB aerosols and their eastward transport life cycle last a few days to weeks (Deng et al., 2008;
Huang et al., 2020; Adam et al., 2021). Thus, significant positive BC AOD anomalies are still observed over the northern
ICP, southwestern China and the Northwest Pacific east of Taiwan in early-April (Fig. 3b). However, the precipitation
anomaly pattern is roughly opposite to that in March, with above-normal precipitation from the northern Bay of Bengal
eastward to the northern SCS and below-normal precipitation over the middle and lower reaches of the Yangtze River (Fig.
3e). Correspondingly, significant anomalous northeasterly wind occurs from the middle and lower reaches of the Yangtze
River toward the northern ICP, acting to reduce the climatological south-westerly wind and the water-vapor transport in
southern China. When mid-April comes, no significant BB aerosol anomalies can be found (Fig. 3c), but the positive
precipitation anomalies still exist over the northern and eastern ICP and the Beibu Gulf, accompanied by anomalous westerly
wind across the Indo-Pacific Ocean and southwesterly wind from the northern tropical Indian Ocean to the northwestern ICP
(Fig. 3f). As no significant anomalies are found in circulation and precipitation after about April 20th, we will focus on the
features in early- to mid-April.
As mentioned above, the March BB aerosols can reduce precipitation over the ICP in March but increase precipitation from
April 1st to around April 20th, indicating that the effects of March BB aerosols on precipitation can last from March to early-
to-mid April, but with opposite effects in the two months. Due to the covariation of aerosols and meteorological fields, it is
hard to determine the causality between BB aerosols over the ICP and atmospheric circulation (and precipitation), especially
using instant observations. Therefore, in the following section, we will use two groups of WRF-Chem experiments to reveal
the physical mechanisms responsible for these relationships.
**4 Numerical modeling results**
**4.1 Evaluation of model results**
Figures 4a–b illustrate the spatial patterns of the observed and modelled rainfall and 850-hPa wind averaged from March 1st
to April 20th, 2010. The TRMM data shows a large rainfall belt extending from the Nanling Mountains to the south of the
Yangtze River (110°–120°E, 23°–30°N) (Fig. 4a), known as spring persistent rainfall in Jiangnan of China (SPRJ). (Note:
Jiangnan is the name in Chinese for the region south of the Yangtze River). In addition, large amounts of precipitation can
also be found over the northwestern ICP region, which is typical orographic precipitation on the windward side of the slope.
The WRF-Chem ensemble-mean rainfall based on six CTRL members (Fig. 4b) shows a spatial pattern consistent with that
in the TRMM, although the model overestimates the convection in the northern tropical Indian Ocean, orographic
precipitation in the northwestern ICP region, and rainfall south of Japan. Similar overestimate tropical convection and
orographic precipitation can be seen in Yang et al. (2022b) using the same model. The atmospheric circulation in East Asia
during early-spring (March 1st–April20th) 2010 is featured by strong easterly winds across the tropical Indo-Pacific Ocean
and southwesterly winds from the Bay of Bengal and SCS to southern China (Fig. 4a). In general, the model can reasonably
capture these observed features of rainfall and circulation.

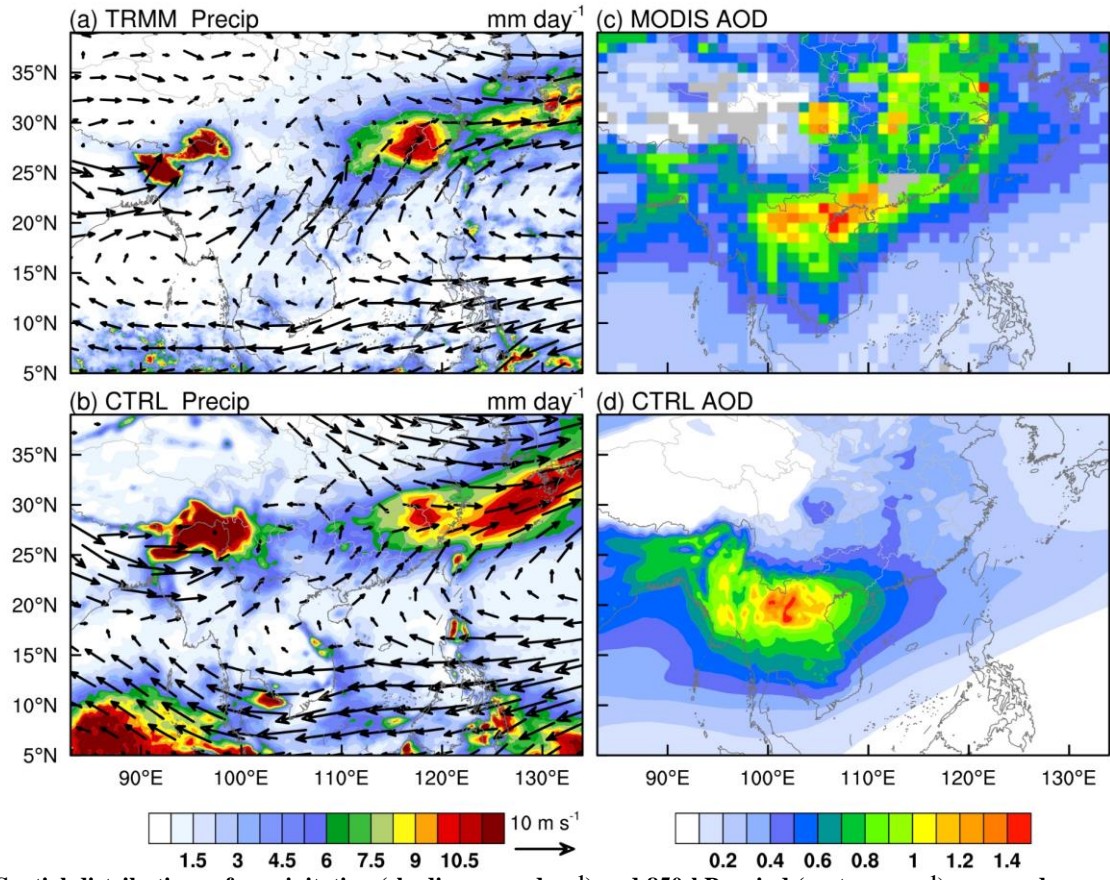

**Figure 4: Spatial distributions of precipitation (shading; mm day⁻¹) and 850-hPa wind (vector; m s⁻¹) averaged over early spring (March 1ˢᵗ to April 20ᵗʰ) of 2010 from (a) observations (TRMM precipitation and ERA-5 wind) and (b) ensemble-mean of WRF-Chem CTRL. (c, d) Sames as (a, b), but for AOD (shading; unitless) from (c) MODIS and (d) ensemble-mean of WRF-Chem CTRL.**

The spatial pattern of modeled AOD is consistent with MODIS satellite retrieval. Figures 4c–d show that the WRF-Chem can capture the observed high aerosol loading over the ICP; however, it underestimates the AOD over eastern China and its coastal regions. The differences between model simulations and satellite data could be attributed to two potential factors. First, the WRF-Chem model does not fully cover the effect of relative humidity on AOD calculation, as increased relative humidity can lead to higher AOD because of aerosol humidification (Myhre et al., 2007). Second, the GOCART AFWA scheme can underestimate the dust aerosol concentration in northwestern China (Zhao et al., 2020), resulting in a lower AOD in northern China. Nevertheless, the WRF-Chem model has a good performance in simulating the BB aerosols over the ICP.

Given this, the ensemble-mean differences between CTRL and BBER (i.e. CTRL minus BBER) are used to examine the effects of BB aerosols and associated physical mechanisms.



## 4.2 Effects of BB aerosols

Figure 5 shows the BB aerosol-induced differences in AOD, rainfall and 850-hPa wind during March and early-to-mid April of 2010. The BB aerosols significantly increased in March due to BB emissions, with a maximum AOD anomaly exceeding 1.2 over the northern ICP (Fig. 5a). The AOD anomaly pattern of AOD agrees well with observations (Fig. 3a). The BB aerosol-induced anomalous circulation exhibits a belt-shaped low-pressure band in the lower troposphere (850 hPa) over Southeast Asia, with two centers located to the east (Hainan Island) and west (coastal southern Myanmar) of the ICP (Fig. 5d). Correspondingly, the precipitation decreased by roughly 13% from the northern Bay of Bengal to southern China. This was probably because the anomalous easterly wind on the northern flank of the low-pressure zone acted to weaken the prevailing southwesterly wind (Fig. 4b), thereby reducing the moisture transport from the Bay of Bengal and SCS. In addition, the precipitation was reduced by about 15% over most of the ICP (Fig. 5d), which was the emission source region. This might be related to the suppressive effect of BB aerosols on local convection (Hodnebrog et al., 2016; Yang et al., 2022b). The largest rainfall reduction occurred in the northwestern ICP, with a maximum exceeding 2 mm day$^{-1}$. The BB aerosol-induced rainfall reduction over the emission source region is consistent with observations (Fig. 3d). Enhanced precipitation occurred in the western and northern SCS, East China Sea, and their coastal regions, under southerly wind anomalies. These simulated changes in rainfall and circulation induced by March BB aerosols agree well with the results based on climate models (Lee et al., 2014; Chavan et al., 2021) and mesoscale weather models (Wang et al., 2021; Yang et al., 2022b).

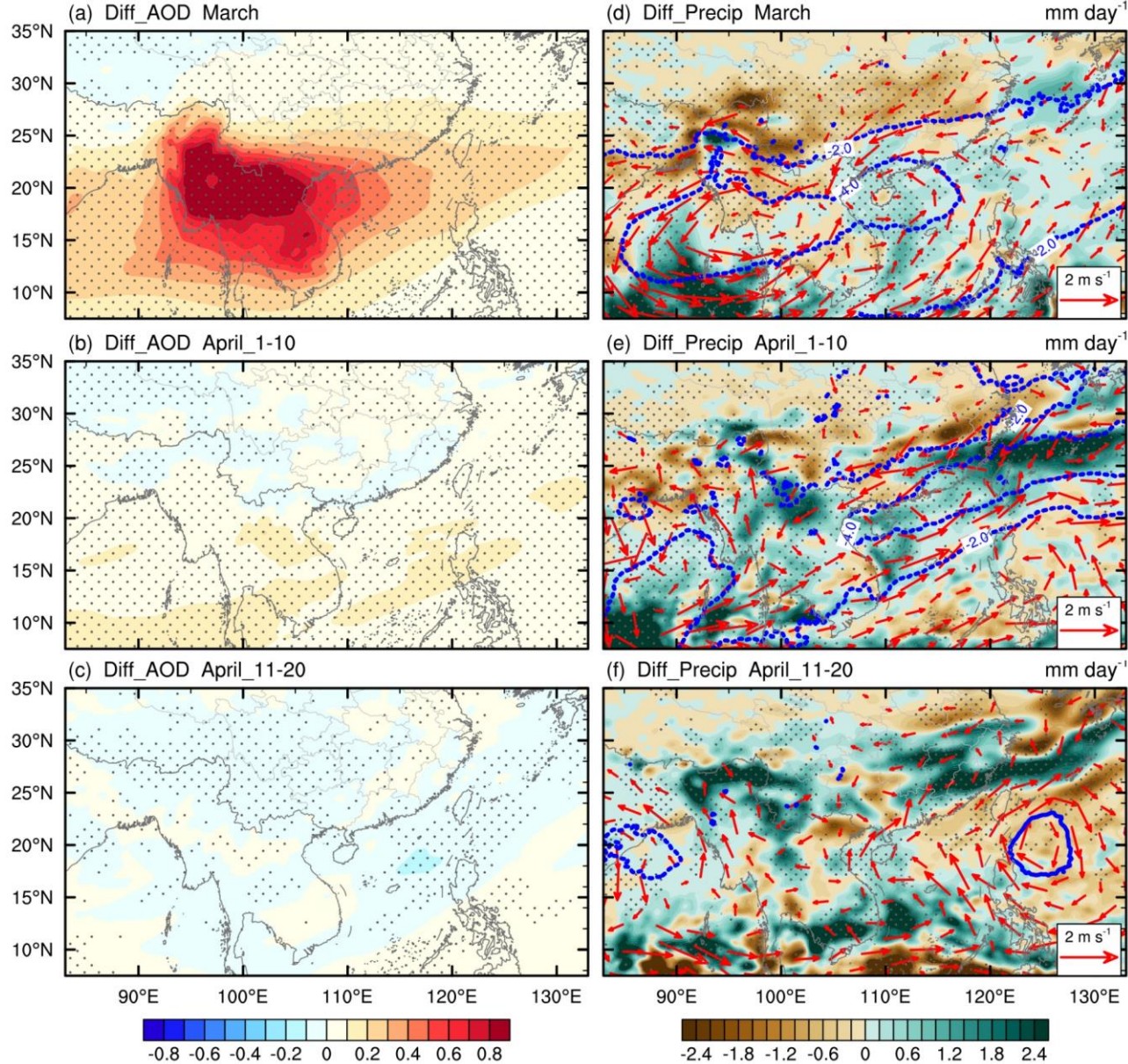

**Figure 5: WRF-Chem-simulated ensemble-mean differences in (a–c) AOD (shading; unitless) and (d–f) precipitation (shading; mm day$^{-1}$), 850-hPa wind (vector; m s$^{-1}$) and geopotential height (blue contours with interval of 2 dagpm; the dashed contours are for negative values and the zero contour is omitted for clarity) between CTRL and BBER (i.e., CTRL minus BBER) during (a, d) March, (b, e) early-April and (c, f) mid- April of 2010. Stippling (red vector) denotes the AOD and precipitation (wind) are statistically significant at the 95% confidence level based on Student's *t*-test.**

As in the observations (Fig. 3b), positive aerosols anomalies due to March BB emissions were still evident (albeit smaller) in early-April (Fig. 5b). The centers of the belt-shaped anomalous low at 850 hPa were located over coastal southern China and the southern Bay of Bengal (Fig. 5e). This indicates that the circulation response to March BB aerosols did not disappear immediately and could last from March to early-April, although it became weak. However, the precipitation promotion due





270 to March BB aerosols dominated over the entire ICP region in early-April, contrary to the rainfall reduction in March.

271 Besides, the SPRJ rainband shifted markedly southward characterized by reduced precipitation in the middle and lower

272 reaches of the Yangtze River and by increased precipitation from coastal Southeast China to the East China Sea. These

273 responses of rainfall and circulation to March BB emissions are similar to those in observations shown in Sect. 3.2. Since

274 aerosol concentration anomalies in April were affected a little by the March BB emissions, the anomalous rainfall in early-

275 April could be potentially caused by the large-scale circulation change.

276 During mid-April, no significant AOD differences appeared over the ICP (Fig. 5c). The BB aerosol-induced belt-shaped

277 850-hPa low-pressure band almost dissipated, with only small cyclonic anomaly wind in the northern Bay of Bengal (Fig. 5f).

278 The anomalous southerly wind in the western ICP transported moisture from the Bay of Bengal to the northern ICP and

279 increased precipitation in the northwestern IPC along the topography on the southeastern side of the Tibetan Plateau. Clearly,

280 the observed circulation and precipitation anomalies in mid-April (Fig. 3f) can also be reproduced in the WRF-Chem model.

281 **4.3 Physical mechanism underlying the BB aerosols-rainfall relationship**

282 **4.3.1 Instant effect**

283 The BB aerosols can significantly change radiative forcing by absorption and scattering of solar radiation, leading to spatial

284 perturbation and redistribution of energy (Chavan et al., 2021). Figures 6a–c shows the BB aerosol-induced changes in net

285 downward SW radiative fluxes at the top of the atmosphere (TOA), in the atmosphere, and at the surface under all-sky

286 conditions in March. BB aerosols can absorb SW radiation and heat up the atmosphere. Thus, positive SW radiation

287 anomalies dominate in the atmosphere over the regions with high BB aerosol loading, with a magnitude of 30–65 W m$^{-2}$

288 from the Bay of Bengal across the ICP to the coastal region of South China and the SCS (Fig. 6b). At the surface, BB

289 aerosols prevent the solar radiation from reaching the surface by scattering and absorption, which causes a surface cooling

290 effect over the high BB aerosol loading regions, as shown in Fig. 6c. The maximum magnitude of the negative SW radiative

291 flux anomalies is about 60 W m$^{-2}$ in the northern ICP. The above BB aerosol-induced SW radiative forcing both in the

292 atmosphere and at the surface are comparable in magnitudes to those found previously (Lin et al., 2014; Pani et al., 2018;

293 Yang et al., 2022b).

**Figure 6: (a–c) Differences (CTRL minus BBER) in all-sky net downward shortwave radiative flux (shading; W m⁻²) (a) at the top of atmosphere (TOA), (b) in the atmosphere (ATM), and (c) at the surface (SFC) in March 2010. (d–f) and (g–i) Same as (a–c), but for clear-sky and cloudy-sky differences, respectively. The purple contours with interval of 0.3 denote AOD differences (CTRL minus BBER). Hatching denotes the radiative effect is statistically significant at the 95 % confidence level based on Student's *t*-test.**

At the TOA, the positive all-sky SW radiative flux anomalies induced by BB aerosols are above 15 W m⁻² over North Vietnam, southern China and the SCS but below 7.5 W m⁻² over the BB emission source region in the northern ICP (Fig. 6a), which is consistent with previous results in both modeling (Lee and Kim, 2010; Dong et al., 2019) and measurement studies (Pani et al., 2016; Pani et al., 2018). Generally, BB aerosols can reflect and scatter more SW radiation back to space compared to BB aerosol-free cases, leading to a weak negative SW radiative forcing at the TOA, as demonstrated in some studies (Lee et al., 2014; Lin et al., 2014; Chavan et al., 2021; Yang et al., 2022b). Nevertheless, absorbing BB aerosols can also switch from exerting a negative to a positive SW radiative effect at the TOA, due to increased underlying cloud coverage or brightness of the underlying layer (Chand et al., 2009; Lu et al., 2018). Thus, in clear-sky conditions (i.e., radiative forcing by aerosols without the cloud-circulation feedback), the TOA SW radiative effect is negative over waters and weak positive over most of the land (Fig. 6d) due to the high surface albedo contrast between those two underlying surfaces, while the strong TOA positive radiative effect over the downstream regions of the BB aerosols' transport is mainly



due to the cloud-circulation feedback. Figures 6g–i show the radiative effects caused by changes in cloud fraction (measured
as the all-sky minus clear-sky radiative effects). Positive radiative effects in cloudy conditions are mostly distributed along
the coastal regions and the ocean waters off southern China and North Vietnam, with a magnitude of 14–28 W m$^{-2}$. Greater
cloud covers occur in these regions (Fig. 7a), which are concentrated in the lower troposphere (i.e., 1000–800 hPa; Fig. 7b).
A previous study demonstrated that the enhancement of low clouds beneath the BB aerosol plume around 3 km over
subtropical East Asia is caused by a synergetic effect of aerosol-cloud-boundary layer interaction with the monsoon (Ding et
al., 2021). In turn, the BB aerosol plume uplifted above the clouds could absorb more solar radiation reflected from the cloud
top, thus reducing the shortwave radiation reflected back to space (Dong et al., 2019). This also means that the increasingly
thick and bright cloud layer underneath the BB aerosol plume would further amplify the direct warming effect in the
atmosphere induced by BB aerosols (Ding et al., 2021), resulting in an increase of atmospheric warming by roughly 15%–20%
(Fig. 6h). The spatial pattern of the net (LW+SW) radiative effect is dominated by the SW radiative effect, because the LW
radiative effect is relatively small. Thus, the LW and net radiative effects are not shown here.

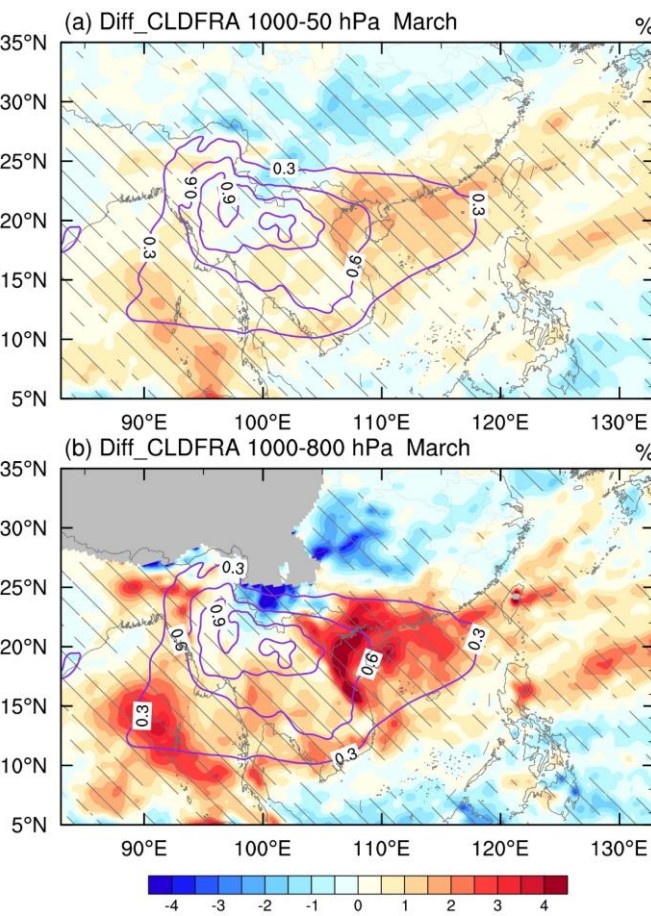

**Figure 7: Differences (CTRL minus BBER) in cloud fraction (shading; %) in the (a) entire atmospheric column (1000–50 hPa) and**
**(b) lower troposphere (1000–800 hPa) in March 2010. The purple contours with interval of 0.3 denote AOD differences. Hatching**
**denotes the cloud fraction change is statistically significant at the 95 % confidence level based on Student's *t*-test.**



BB aerosols can dramatically alter the horizontal and vertical distribution of atmospheric temperatures through their
radiative effects. Figure 8 shows the spatial pattern of BB aerosol-induced temperature changes from surface to 500 hPa in
March 2010. Due to the surface cooling effect of BB aerosols, the surface temperature was reduced by up to 1.6K in the ICP,
and the cooling could reach up to 850 hPa (Figs. 8a–b). The BB aerosol-induced warming at 700 hPa can be widely found
from the Bay of Bengal across the ICP, SCS and southern China to the East China Sea, with a magnitude between 0.4 and
2.0K (Fig. 8c); and such a warming pattern generally follows the AOD anomaly pattern. As a result, the BB aerosol-induced
surface cooling and 700-hPa warming acted to increase the low-level atmospheric stability. Besides, a weak atmospheric
cooling effect was found in the mid troposphere (500 hPa) over the ICP (Fig. 8d).

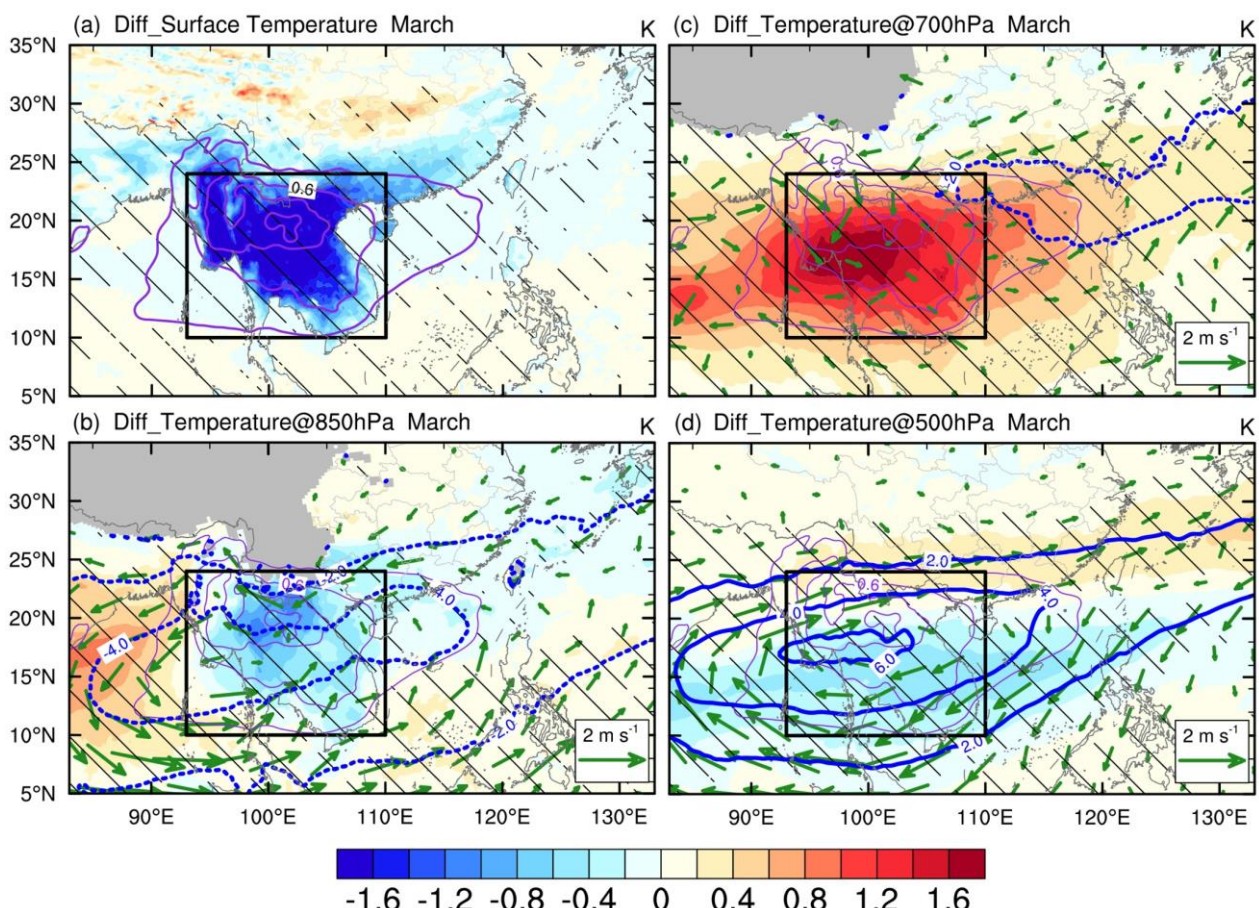

**Figure 8: Differences (CTRL minus BBER) in (a) surface temperature (shading; K), (b–d) horizontal wind (vector; m s$^{-1}$),**
**geopotential height (thick blue contours with interval of 2 dagpm; the dashed contours are for negative values and the zero**
**contour is omitted for clarity), and temperature (shading; K) at (b) 850 hPa, (c) 700 hPa, and (d) 500 hPa in March 2010. Purple**
**contours with interval of 0.3 denote AOD differences. The hatching and green vectors denote temperature and wind changes are**
**statistically significant at the 95 % confidence level, respectively, based on Student's *t*-test. The black box outlines the main**
**Indochina Peninsula (ICP; 93 º–110 ºE, 10 º–24 ºN).**
To better explain such "cooling-warming-cooling" vertical temperature changes from the lower to upper troposphere, we
show the vertical profiles of changes in area-averaged atmospheric heating source in the ICP (93 º–110 ºE, 10 º–24 ºN; black





box in Fig. 8) during March (Fig. 9a). As expected, SW radiative forcing was the major factor contributing to the
atmospheric heating, which was the strongest (exceeding 1.0K day$^{-1}$) near 650 hPa and diminished to zero near 400 hPa.
Note that the height of the SW heating did not coincide with that of the BC mass concentration maximum, partially due to
the amplification heating effect caused by the increased low-cloud underneath the BB smoke plume (Fig. 9b). The surface
cooling caused by the solar flux reduction tends to decrease surface evapotranspiration, and reduce sensible and latent heat
fluxes (Andreae et al., 2004; Feingold et al., 2005; Huang et al., 2016b). As a result, the PBL processes dominate the cooling
effect in the lower troposphere (1000–700 hPa). This can also explain why the PBL cooling was weaker over the ocean than
over land (Figs. 8a–b), as the surface fluxes over the ocean were much less variable (Feingold et al., 2005). The latent heat
shows a weak warming effect from ~950 to 750 hPa, which can translate to promoting cloud formation by large-scale
condensation and even moist convection. As shown in Figs. 7 and 9b, the increase in low clouds over the Beibu Gulf was
concentrated below 850 hPa, while that over the southern ICP was at 850–750 hPa. Additionally, the latent heating also
displayed a weak cooling effect at 700–500-hPa because of the reduced clouds in this layer via the cloud burn-off effect of
BC (the semi-direct effect). The LW radiative forcing heating contributed to the atmospheric cooling from the surface to
about 400 hPa. The net atmospheric heating (i.e., the sum of SW, LW, PBL, and latent heat), induced by BB aerosols
generally exhibited a cooling effect below 850 hPa and a warming effect at 850–400 hPa. As a result, the colder temperature
anomalies occurred from the surface to 800 hPa with a minimum reaching -0.76K, while warmer anomalies with a maximum
greater than 1K were around 800–550 hPa (Fig. 9a). These temperature anomalies can markedly increase the atmospheric
stability in the lower troposphere, leading to a more unstable mid troposphere.

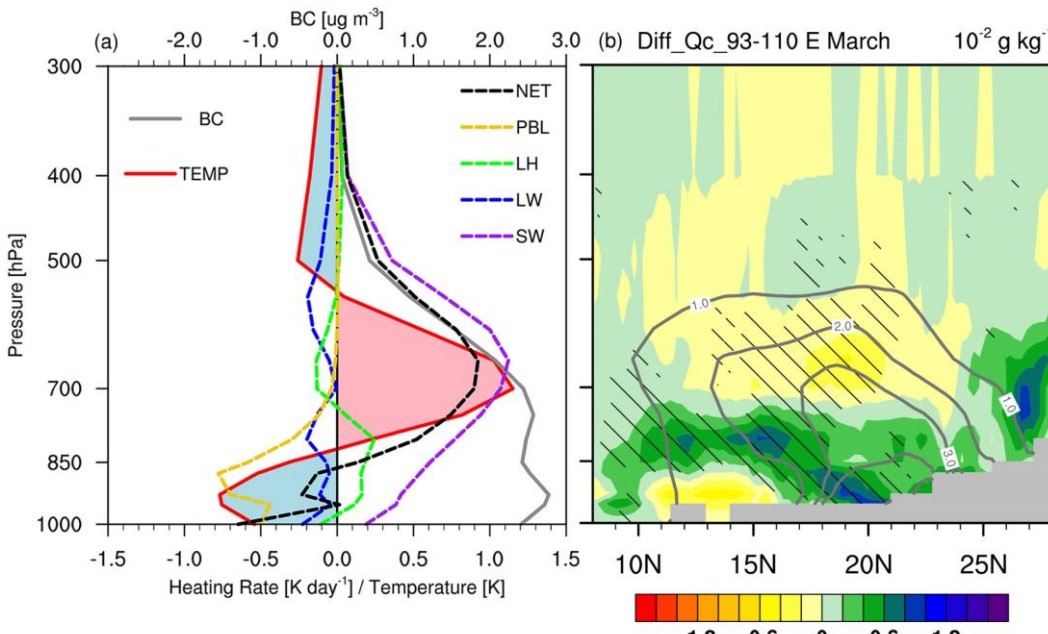

**Figure 9: (a) Vertical profiles of differences (CTRL minus BBER) in temperature (solid red line; K), BC mass concentration (solid**
**grey line; ug m$^{-3}$), and atmospheric heating rates (dashed line; K day$^{-1}$) averaged over ICP (93 °–110 °E, 10 °–24 °N; as outlined in Fig.**
**8) in March 2010. Here, atmospheric heating rates include shortwave (SW) and longwave (LW) radiation heating, latent heating**





**(LH; i.e., heating from microphysics and cumulus scheme), and heating from planetary boundary layer (PBL) scheme. Net heating rate (NET) = SW + LW + LH + PBL. (b) Vertical cross-sections of differences (CTRL minus BBER) in cloud water-vapor content (shading; $10^{-2}$ g kg$^{-1}$), BC mass concentration (solid grey contours with interval of 1.0 ug m$^{-3}$) averaged over 93°–110°E in March 2010. Hatching denotes changes in cloud water-vapor content are statistically significant at the 95 % confidence level based on Student's $t$-test.**

The BB aerosol-induced maximum net heating in the troposphere could reach up to 0.9K day$^{-1}$ (Fig. 9a), which was able to force anomalous atmospheric circulation. As suggested previously (Hoskins, 1991; Wu and Liu, 2000), the atmospheric response to an external diabatic heating can generate upward motion in the heating layer, cyclonic circulation in the lower atmosphere and anticyclonic circulation in the upper troposphere. These anomalous circulations can be clearly seen in our simulation results shown in Figs. 8b–d. Furthermore, subject to atmospheric thermal adaptation (Wu and Liu, 2000; Liu et al., 2001), the "overshooting" air parcel induced by the inertial ascent from below the heating layer kept a constant potential temperature, forming the cold anticyclonic circulation to the northwest of the heat source in the upper troposphere (Figs. 10a – b). Accordingly, anomalous northerly (southerly) winds across the heating region in the upper (lower) troposphere (Fig. 10a) developed to balance the Coriolis force (Liu et al., 2001). To the north of the BB aerosol heating region (22°–26°N), the negative meridional diabatic heating gradient produced a negative vorticity forcing and a secondary circulation at the upper level (Figs. 10a, c). The BB aerosol-induced two-cell structure meridional circulation is quite similar to the results in Lee and Kim (2010) and Yang et al. (2022b). The sinking motion in the northern branch is consistent with the maximum precipitation anomaly in Fig. 5d. The anomalous northwesterly flow on the northern flank of the cyclonic circulation in the lower troposphere substantially weakened the water vapor transported from the Bay of Bengal to the northern ICP and southern China (20°–30°N; also see Fig. 10a). However, more water vapor was lifted up from the Bay of Bengal and SCS into the mid troposphere via the Ekman pumping (Fig. 10b), which was partly transported to the central and southern ICP by anomalous southerly wind in the southern branch (Figs. 10a–b). Interestingly, precipitation was reduced in the central and southern ICP by the BB aerosols, despite of the favorable water-vapor condition (Fig. 5d). This is because the increased atmospheric stability in the low-troposphere caused by the BB aerosols greatly enhanced the convection inhibition energy (CIN) (Fig. 10d), indicative of a higher threshold for the energy required to trigger convection (Mapes, 2000). As a result, the reduction of the local convective rainfall dominated the change in precipitation over the ICP (Fig. 11a), while large-scale (stratiform) precipitation presented a minor increase (Fig. 11b). The effects of BB aerosol-induced suppression of convective precipitation and mild enhancement of large-scale precipitation over the northern ICP are consistent with the modelling results of Wang et al. (2021).

**Figure 10: (a–c)** Vertical cross-sections of differences (CTRL minus BBER) in temperature (shading; K), geopotential height (blue contours with interval of 1.2 dagpm; the dashed contours are for negative values and the zero contour is omitted for clarity ), and water-vapor content (green contours with interval of 0.3 g kg⁻¹ for positive values and of 0.1 g kg⁻¹ for negative values, and the zero contour is omitted for clarity) averaged over (a) 93 º–110 ºE , (b)10 º–24 ºN, and (c) 110 º–120 ºE in March 2010, together with (a, c) meridional [or (b) zonal], vertical velocity (vector; m s⁻¹ and 10⁻² m s⁻¹, respectively) and BC mass concentration (solid grey contours with interval of 1.0 ug m⁻³). **(d)** Same as (b), but for convective inhibition (CIN; shading; J kg⁻¹). Hatching and vectors denote the shaded field and wind changes are statistically significant at the 95 % confidence level, respectively, based on Student's *t*-test.


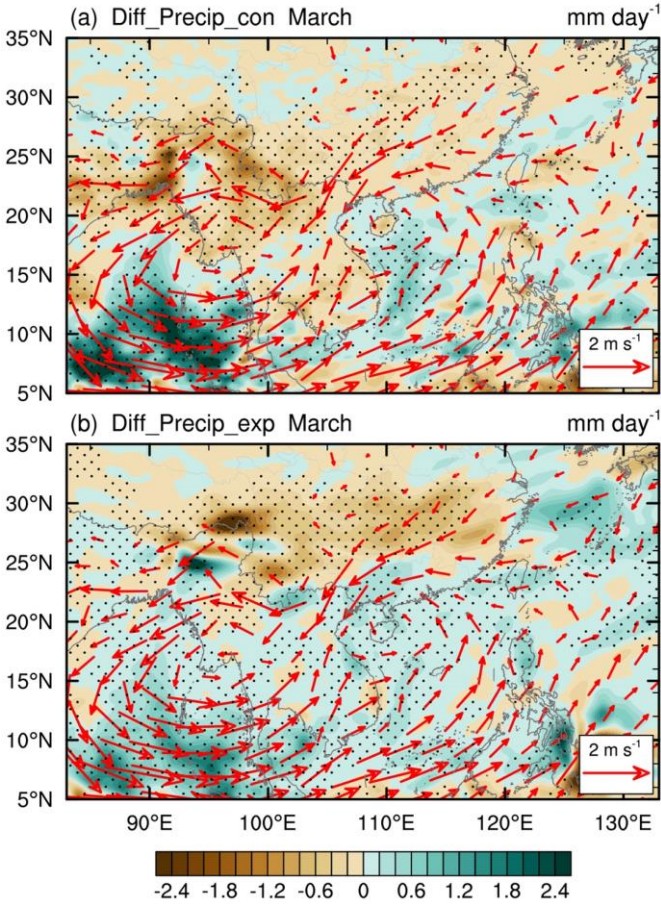

**Figure 11: Differences (CTRL minus BBER) in (a) convective precipitation and (b) non-convective precipitation (shading; mm**
**day⁻¹) in March 2010, together with 850-hPa wind difference (vector; m s⁻¹). Stippling and red vector denote precipitation and**
**wind are statistically significant at the 95% confidence level, respectively, based on Student's *t*-test.**
For the SCS and its adjacent coastal water region (110 °–120 °E), the PBL cooling was quite weak (Fig. 10c), resulting in little
CIN change in the lower layers (Fig. 10d). Therefore, relatively favorable water-vapor conditions led to moderately
enhanced precipitation (Fig. 5d). This is similar to the "elevated heat pump" (EHP) effect proposed by Lau et al. (2006),
which hypothesized that the absorbing aerosols (dust and BC) stacked up on the southern slope of the Tibetan Plateau can
heat up the mid-to-upper troposphere, leading to an earlier onset of the Indian summer monsoon and increased monsoon
rainfall. Note that in our case the updraft caused by the low-level (700-hPa) heating only reached 500 hPa, leading to an
invigoration of shallow convection, which differs from the original "EHP" effect with a high-level (500-hPa) heating and a
resultant ascent air flow reaching 200 hPa.
**4.3.2 Delayed effect**
Compared to the instant effect, the delayed effect in the subsequent April should be closely related to the atmospheric
circulation adjustment, as there were a few BB aerosols left from March. During the subsequent early-April, the anomalous
vertical temperature structure still persisted with a maximum warming of 0.4K at 700 hPa and cooling of -0.6K at 925 hPa
(Figs. 12a, c). Without the strong heating from the BB aerosols (Fig. 12c), the 850-hPa anomalous low over the ICP became
weaker and split into a double-center system (Fig. 12b). This would increase moisture over the northern ICP and northern
SCS by southerly anomalies, which facilitated precipitation over the northern ICP, southern China and the northern SCS
(Figs. 12b, d and Fig. 5e).

**Figure 12: (a) Vertical cross-sections of differences (CTRL minus BBER) in temperature (shading; K), geopotential height (blue**
**contours with interval of 1.2 dagpm; the dashed contours are for negative values, and the zero contour is omitted for clarity), and**
**water-vapor content (green contours with interval of 0.3 g kg⁻¹ for positive values and of 0.1 g kg⁻¹ for negative values, and the zero**
**contour is omitted for clarity), together with meridional and vertical velocity (vector; m s⁻¹ and 10⁻² m s⁻¹, respectively) and BC**
**mass concentration (solid grey contours with interval of 0.2 ug m⁻³) averaged over 93 º–110 ℉. (b) Differences (CTRL minus BBER)**
**in 850-hPa wind (vector; m s⁻¹), geopotential height (blue contours with interval of 1.2 dagpm), water-vapor content (green**
**contours with interval of 1.0 g kg⁻¹ for positive values and of 0.2 g kg⁻¹ for negative values, and the zero contour is omitted for**



**clarity), and temperature (shading; K). (c) Vertical profiles of differences (CTRL minus BBER) in temperature (solid red line; K),**
**BC mass concentration (solid grey line; ug m⁻³), and atmospheric heating rates (dashed lines; K day⁻¹) averaged over the ICP**
**[black box in (b)]. (d) Same as (b), but at 700 hPa. Hatching and vector denote the shaded field and wind changes are statistically**
**significant at the 95 % confidence level, respectively, based on Student's *t*-test. All of them are averaged over April 1ˢᵗ–10ᵗʰ, 2010**
**(i.e., early-April).**
As analyzed in Sect. 4.3.1, the rainfall reduction over the ICP in March induced by BB aerosols resulted from competition
between convection suppression by the stabilized atmosphere and favorable water vapor-conditions by large-scale
circulation response. For the delayed effect in early-April, favorable water-vapor conditions due to atmospheric circulation
adjustments increased significantly, as the low-level anomalous low weakened and the monsoon advanced. On the other
hand, the convective instability above 850 hPa was significantly enhanced under the influence of water vapor (Fig. 13c),
although the BB aerosol-induced anomalous vertical temperature structure remained. In other words, both conditions were
conducive to the precipitation over the ICP in the early-April. Thus, the delayed effect acted to promote precipitation over
the ICP, in contrast to inhibiting precipitation by the instant effects. In turn, the increased condensation heating associated
with increased rainfall dominated the upper-air diabatic heating (Fig. 12c) via positive feedback. The adjustment in the net
maximum heating layer height also led to an anomalous cyclonic circulation at 700 hPa (Fig. 12d). Due to the memory of the
soil, the reduction in land surface variables such as soil temperature, soil moisture and surface evaporation can last until this
period and keep the cooling effect through the PBL process (Fig. 12c). Then, all these factors acted to maintain the
anomalous vertical structure of PBL cooling, upper-air warming and the anomalous circulation, so that the preceding
atmospheric responses would not disappear immediately.

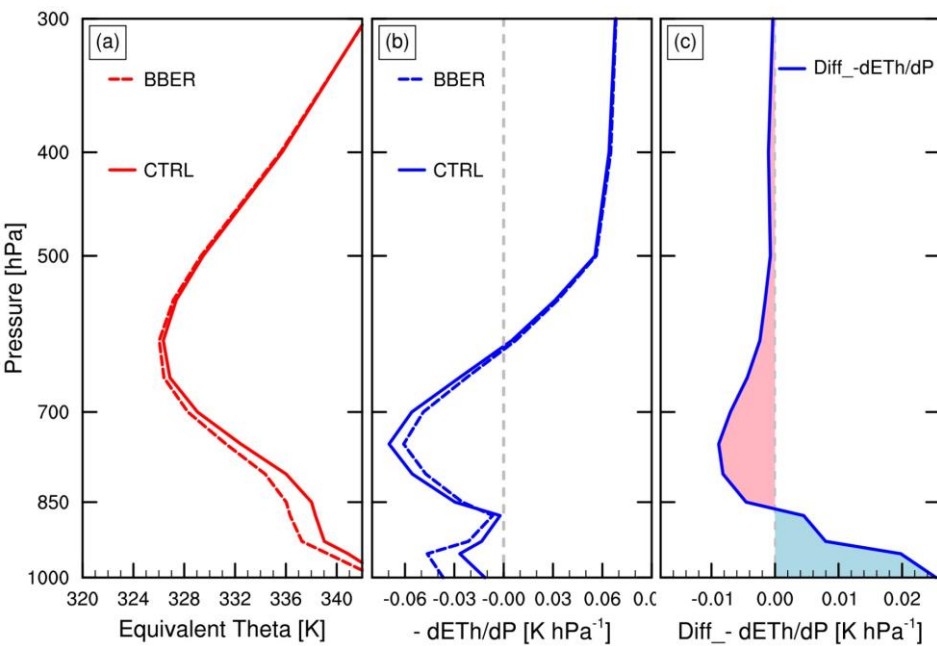


**Figure 13: Vertical profiles of (a) equivalent potential temperature ($\theta_e$; red line; K) and (b) convective stability ($-\frac{\partial \theta_e}{\partial P}$; blue line; K**
**hPa⁻¹) averaged over the ICP (as outlined in Fig. 12b) during April 1ˢᵗ-10ᵗʰ, 2010. The solid and dash lines are for CTRL and**
**BBER, respectively. (c) Differences (CTRL minus BBER) in the convective stability (blue line; K hPa⁻¹).**





Without the anomalous heating from the BB aerosols during the mid-April, the anomalous vertical temperature structure was
barely seen over the ICP (Figs. 14a, c). Meanwhile, as the 850-hPa anomalous low further dissipated, anomalous southerly
wind transported more water vapor from the Bay of Bengal directly northward to the northwestern ICP (Fig. 14b). The moist
airflows were then lifted by the southeastern Tibetan Plateau and thus converged and cooled, which enhanced orographic
precipitation (Fig. 5f). Although the BB aerosol-induced anomalous low nearly disappeared over coastal Southeast China
during the mid-April, the anomalous meridional circulation accompanied by enhanced precipitation over southern China (Fig.
5f) could be sustained through the feedback from the increased condensation heating.
**Figure 14: Same as Fig. 12, but the field are averaged over April 11th-20th, 2010 (i.e., mid-April).**





**5 Conclusions and discussion**
Large amounts of absorbing aerosols are injected into the atmosphere by extensive BB activities over the ICP during March,
which can significantly affect the regional climate. Using observation data and the WRF-Chem model, we investigate the
instant and delayed effects of the BB aerosols over the ICP in March on the regional circulation and precipitation in early-
spring. The main conclusions are summarized below.
The observations show that March BB aerosols are negatively correlated with the rainfall over the ICP, while such a
correlation shifts to be positive in early- and mid-April, which is well captured by the WRF-Chem model. The simulation
results reveal that BB aerosols emitted from the northern ICP trap a substantial proportion of solar radiation in the low-to-
mid troposphere and decrease incoming solar radiation at the surface, followed by reduced surface heat fluxes associated
with PBL processes. The energy perturbation leads to temperature changes in surface and lower tropospheric (1000–850-hPa)
cooling and lower-to-mid tropospheric (850–400-hPa) heating. Thus, the low atmosphere is stabilized and CIN is markedly
intensified at 850–700 hPa, which acts to suppress local convective rainfall. The BB aerosol-induced heating in the low-to-
mid troposphere can also cause an anomalous low-pressure system in the lower troposphere extending from the central Bay
of Bengal across the ICP to the northern SCS. This is accompanied by a two-cell structure meridional circulation with rising
motion over the ICP and two strong downward motions in the near-equatorial regions and the latitudes of 25 °–30 °N. Over
the ICP, the anomalous low in the lower troposphere tends to increase the mid-tropospheric moisture from the Bay of Bengal
and SCS via moisture advection and Ekman pumping. On the southern flank of this anomalous low, the southerly wind
conveys more water vapor to the ICP, causing a minor increase in large-scale precipitation. Thus, the BB aerosol-induced
rainfall suppression in the ICP during March is a result of competition between the responses of local atmospheric stability
and large-scale circulation to absorbing aerosols. For the SPRJ region, the anomalous northeasterly wind on the northern
flank of the anomalous low would decrease the prevailing southwesterly wind and moisture transport, which is conducive to
suppress the rainfall over these regions. Meanwhile, the sinking motion in the northern branch of anomalous two-cell
structure meridional circulation induced by BB aerosols would also help reduce the precipitation there. Over the SCS, the
moderate precipitation increase is due to favorable water-vapor conditions, while the CIN increases very little because of the
insignificant PBL cooling, which is caused by the underlying water surface.
During early-April, the anomalous belt-shaped low-pressure weakens and fragments into a double-center system, owing to a
few BB aerosols remaining in March and the corresponding reduction in BB aerosol-induced atmospheric heating. Over the
ICP, although the anomalous low weakens due to lack of strong heating from the BB aerosols, it can still transport sufficient
moisture from the Bay of Bengal as the monsoon advances. On the other hand, the convective instability above 850 hPa is
enhanced under the influence of water vapor, although the vertical temperature anomaly structure remains. As a result, the
effects of March BB aerosols on precipitation over the ICP shift from suppression in March to enhancement in early- and
mid-April. In turn, the increased condensation heating associated with increased rainfall dominates the diabatic heating and
sustains the anomalous circulation and vertical temperature structure via positive feedback. In mid-April, without any





anomalies directly related to BB aerosol-induced heating, the anomalous vertical temperature structure and low pressure in
the lower troposphere nearly disappear, and only enhanced rainfall over the northwestern ICP and southern China can be
seen due to the condensation heating.
Recently, Yang et al. (2022b) investigated the effects of BB aerosols from the ICP during the whole emission season (March
$1^{st}$–April $17^{th}$, 2010). In this study, we further discuss the instant and delayed effects in the peak BB emission month of
March. The instant effect of March BB aerosols on the atmospheric circulation is consistent with the results of Yang et al.
(2022b). Interestingly, Yang et al. (2022b) noted that the April BB aerosols could significantly enhance the heavy rain events
over the southern coast of southern China, while we show that the BB aerosol perturbation in March can induce a delayed
increase in April precipitation over the same region. For the precipitation decrease over southern China, in addition to the
cyclonic anomalies that reduce water vapor transport as stated by Yang et al. (2022b), we find that the sinking motion in the
anomalous vertical meridional circulation induced by BB aerosol's heating also plays a role. Using an AGCM, Lee et al.
(2014) suggested that the indirect effect is the main contributor to the BB aerosol-induced precipitation suppression over the
ICP. In contrast, Ding et al. (2021) demonstrated that the indirect effects of BB aerosols play a less significant role in the
low-cloud enhancement over subtropical Asia. Although both direct and indirect effects of aerosols are included in our
experiments, we focus on the aerosol-radiation interaction (i.e., direct or semi-direct effect). The role of indirect effects
needs to be investigated by setting up experiments with and without indirect effects in further.
Note that the modeling results in this study focuses only on the year of 2010, during which the AOD magnitude in March
was greater than its climatology by about 1.7 standard deviations. The effects of aerosols on precipitation in the model (Figs.
5d–f) are not fully consistent with observations (Figs. 3d–f), especially for the delayed effects (Figs. 3e, f and Figs. 5e, f).
Due to the fact that the response patterns of large-scale circulation and precipitation to BB aerosols largely depend on both
aerosols and meteorological conditions. Thus, multiyear simulations are needed to assess the robustness of our results on a
longer time scale. In addition, uncertainty may also exist in the simulation. For instance, the overestimate of convective
rainfall in the tropical Bay of Bengal and orographic precipitation in the southeastern Tibetan Plateau might introduce some
uncertainty in the response of large-scale circulation to BB aerosols, which is strongly related to the cumulus convection
parameterization scheme and the topographic complexity (Ma and Tan, 2009; Li et al., 2022). Therefore, further experiments
at convection-resolved resolution need to be conducted to reduce such uncertainty.
**Code and Data availability.**
The source codes of WRF-Chem model are available at https://www2.mmm.ucar.edu/wrf/users/download/get_source.html.
The FNL data are available at https://rda.ucar.edu/datasets/ds083.2/. The BB emission data of FINN version 1.5 are available
at https://www.acom.ucar.edu/Data/fire/. The MEIC and MIX anthropogenic emissions are available at
http://meicmodel.org/?page_id=541&lang=en. The ERA-5 Reanalysis data are available at
https://cds.climate.copernicus.eu/cdsapp#!/search?type=dataset. MERRA-2 aerosol reanalysis data, OMI AI and TRMM




precipitation are available at https://disc.gsfc.nasa.gov/datasets. The MODIS AOD are available at
https://ladsweb.modaps.eosdis.nasa.gov/missions-and-measurements/products/MOD08_M3. The BB emission data of
GFEDv4 are available at https://daac.ornl.gov/VEGETATION/guides/fire_emissions_v4_R1.html.
**Author contributions.**
HX and AZ conceptualized the research goals and aims. SH and AZ ran the simulations. AZ performed the data analysis and
visualized the results. AZ, HX, JD, and JM wrote the initial draft.
**Acknowledgments**
This work is jointly supported by the National Natural Science Foundation of China (41975106 and 42192562). We
acknowledge the High Performance Computing Center of Nanjing University of Information Science & Technology for their
support of this work. We also thank all the corresponding institutions for providing their data for this study.
**Competing interests**
The authors declare that they have no conflict of interest.
**Financial support**
This research has been supported by the National Natural Science Foundation of China (41975106 and 42192562).

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
