# Peer review of "Instant and delayed effects of march biomass burning aerosols over the Indochina Peninsula"

_Atmospheric Chemistry and Physics, 2022_

## Referee Comment (RC2)

Reviewer Comment on "Instant and delayed effects of March biomass burning aerosols over the Indochina Peninsula" by Zhu et al.

This paper uses WRF-Chem model to explore the impact of biomass burning (BB) on atmospheric circulation and precipitation during the peak biomass burning season (March) in the Indochina Peninsula (ICP). Authors utilize observations to show that March BB aerosols can reduce precipitation over the ICP in March but increase precipitation from April 1-20, indicating the long-lasting effects of March BB aerosols on precipitation, but with opposite effects in the two months. However, it is hard to determine the causality between BB aerosols over the ICP and atmospheric circulation (and precipitation), just from observations. Therefore, two groups of WRF-Chem experiments: with control (CTRL) and sensitivity (BBER) model scenarios, were performed to discern the mechanisms responsible for these feedbacks of BB aerosols on precipitation. To discern the feedback effects.

The paper is very well-written with brevity and high-quality visualization of all results. The manuscript should be published once the following comments are addressed:

**Specific comments:**

- 1) Section 4.1: Evaluation of model results
  - For comparison of CTRL AOD and Precipitation with MODIS and TRMM observations respectively, if possible, please provide some domain-wide statistical difference metrics (such as mean bias, mean error, RMSE, correlation, etc.). This will be helpful to quantify the predictive capability of the default model for the ICP region in this study period.
- 2) Besides, discerning the direct and indirect effects via sensitivity experiments with and without direct or indirect effects, doing an HYPLIT trajectory analysis to look at BB emissions trajectories in the modeling domain for the March-April period, if possible is encouraged. Trajectories of air mass relative to the black box that outlines the main Indochina Peninsula (ICP; 93°–110°E, 10°–24°N), may aid the authors' current inferences more in explaining the opposite impacts in March vs April.
- Adding maybe a supplemental figure on the BB emissions for the ICP region focusing on the March-April season might be helpful in explaining the instant vs delayed impacts of BB burning aerosols in ICP regions' atmospheric circulation and precipitation patterns.

---

## Author Comment (AC1)

**Responses to Anonymous Referee #1**

This paper used WRF-Chem model to explore the feedback mechanisms of BB aerosols-climate interactions during March biomass burning in Indochina Peninsula. They reported that BB released aerosols could inhibit precipitation in ICP, showing instant and lag effects on local and regional rainfall. To discern the feedback effects, authors conducted two groups of simulations with control (CTRL) and sensitivity (BBER) model scenario experiments. They then compared the results from CTRL and BBER to quantify the BB effects.

**Responses:** We greatly appreciate these comments and suggestions. The manuscript has been improved by considering these comments and suggestions. Our detail responses are given point by point below in blue. The revised text is highlighted in red.

The paper is well-written and result analysis makes sense. My concern is that WRF-Chem modeling system fully coupled with two-way interaction offers unique option to examine the association between aerosols and meteorology. The feedback configuration takes into account both direct and indirect aerosol effects, thereby to provide better interpretations to the feedback effects on meteorology by turning on and off the feedback configuration. Authors' modeling investigation rely largely on the selection of BB emission reduction or enhancement in the sensitivity experiment.

**Responses:** As mentioned by the reviewer, aerosol climate effects can also be examined using the WRF-Chem model by turning on and off the feedback configuration, as in previous studies (e.g., Ding et al., 2021; Wang et al., 2021). However, emission perturbations have also been widely used in BB aerosol climate effect modelling studies apart from switching feedback processes. For instance, both Lee and Wang (2020) and Takeishi and Wang (2022) investigated the impacts of BB aerosols over the Maritime Continent based on their WRF-Chem experiments with and without BB emissions. Similarly, Lee et al. (2014), Dong et al. (2019) and Chavan et al. (2021) used this approach to study the BB aerosols' impacts on climate over the Indochina Peninsula (ICP). Thornhill et al. (2018) compared two simulations with the highest and lowest observed BB emissions in South America during 1997–2011. Yang et al. (2022) characterized the effects of BB aerosols by using the difference between the simulation with double (×2) BB emission and that with controlled (×1) BB emission. Liu et al.

(2020a) designed a complex series of experiments that included both four BB emission scenarios and turning-on and -off aerosol radiation feedback, in order to examine the relative importance of aerosol-cloud interactions (ACIs) and aerosol-radiation interactions (ARIs) over the Amazon.

Nevertheless, as concerned by the reviewer, we also ran an additional case same as CTRL, except that aerosol-climate feedbacks (both ARIs and ACIs) were turned off (labelled as Non-Feedback; only one member starting on February 25[th] 2010). As shown in Figure R1, the anomalous patterns of precipitation and 850-hPa wind from these two methods are quite similar, confirming that our method of reducing BB emission in this study is reliable. Note that the precipitation and 850-hPa wind anomalies from the Non-Feedback run are somewhat greater than those from BBER. This may be because 100% of the total BB aerosols are considered in the Non-Feedback run (their ACIs and ARIs were turned off), while only 85% of the total BB aerosols are considered in the differences between CTRL and BBER (i.e., CTRL minus BBER). It should also be noted that we only turned off the ACIs and ARIs from all anthropogenic aerosols (including BB aerosols) in the Non-feedback run, but the similar anomaly patterns from the BBER and Non-Feedback runs at least confirm that our results are reliable and further suggest that the effect of BB aerosols dominates in the BB season over the ICP, while other anthropogenic aerosols might only play a minor role. Overall, we believe that results should be similar based on these two simulating approaches.

To simplify the process, we reduced the BB emission to 15% in the BBER run because the BB emission over the ICP in 2001, the year with the lowest BB emission during 1998–2019, is about one-sixth of that in 2010. This would be more realistic. Nevertheless, the BB emission reduction rate in the sensitivity experiment may also affect our results. Thus, we also ran another test with half BB emission (Half; only one member starting on February 25th 2010). As shown in Figure R2, the differences in precipitation and 850-hPa wind of the two BB emission reduction scenarios with respect to CTRL are quite similar in pattern but with different magnitude. For example, the rainfall anomalies are 72.73%, 36.15% and 31.50% of that from BBER in March, early- and mid-April, respectively. This indicates that our qualitative conclusions are robust. We understand that the responses of atmospheric circulation and precipitation to aerosols are not linear with increasing aerosol loading (Liu et al., 2020a). Thus, we plan to quantify the climate responses to BB emission by designing different emission

scenarios over the ICP in the future.

[Figure]

**Figure R1: Comparison of model simulations with different model settings. (a) The differences in precipitation (shading; mm day⁻¹) and 850-hPa wind (vector; m s⁻¹) between CTRL and Non-Feedback (i.e., CTRL minus Non-Feedback) during March of 2010. (b) Same as (a), but for the differences between CTRL and BBER (i.e., CTRL minus BBER). Note that both CTRL and BBER runs here contain only one member of the ensemble runs, which started on February 25th 2010.**

[Figure]

**Figure R2: Comparison of model simulations with different BB emission reduction scenarios. (a–c) The differences in precipitation (shading; mm day−1) and 850-hPa wind (vector; m s−1) between CTRL and BBER (i.e., CTRL minus BBER) during (a) March, (b) early-April and (c) mid-April of 2010. (d-f) Same as (a–c) but for the differences between CTRL and Half (i.e., CTRL minus Half). The black box outlines the main region of precipitation anomalies in the ICP (92 º–108 ºE, 12 º–27 ºN), and the regional mean is given at the top-left corner of each panel. Note that the CTRL and BBER runs here have only one member of the ensemble runs, which started on February 25th 2010.**

To state more clearly about our experimental design, we have revised the relevant statements around Page-9, Lines 167–170 in the revised manuscript to the following: "*We chose the year of 2010 for modeling because the BB emission in 2010 was greater than its climatology by about 1.7 standard deviations. We reduced BB emission to 15%*

*as the sensitivity experiment in this study, because the March BB emission over the ICP in 2001, the year with the lowest BB emission during 1998–2019, is roughly 15% of that in 2010. It would be more realistic to investigate the effects of BB aerosols on atmospheric circulation and precipitation on the interannual timescale.*"

In response to this comment, the following text has been added to the conclusions and discussion section (Page-29, Lines 539–550): "*It is worth noting that this study examines the BB aerosol climate effects using the model by reducing BB emission, while another method is commonly used, namely, by turning on and off the aerosol climate feedback configuration (e.g., Ding et al., 2021; Wang et al., 2021). We have done a simple verification, and found that the results obtained by the two methods are similar (Fig. S2). Additionally, although some quantitative results can be derived in this study, such as a 12.94(±4.22)% reduction (the value after "±" is a single standard deviation, hereafter the same) in rainfall in the ICP (92°–108ºE, 12°–27ºN) due to March BB aerosols' instant effect, and 15.40(±5.11)% and 13.93(±5.65)% enhancements from the delayed effect in early- and mid-April, respectively, these quantitative results would rely on the BB emission reduction rate in the sensitivity experiment. A supplementary sensitivity test with 50% BB emission showed that the anomalous patterns of 850-hPa wind and rainfall are quite similar to those from BBER, but the rainfall anomalies are 72.73%, 36.15% and 31.50% of those from BBER in March, early- and mid-April, respectively (Fig. S3), indicating our qualitative conclusions are robust. As for quantitative results, this study is based on preliminary analysis; more experiments with different BB emission scenarios need to be designed to obtain more precise results in the future.*"

Can authors provide a BB emission map in Indochina Peninsula? The BB emission in this region seems far less than that in India and sub-Saharan Africa.

**Responses:** We have replaced Figure 2 by the one overlaid with March BC emission from BB as well as the time series of the BB emissions (BC, OC and SO2) for both CTRL and BBER runs (Page 7 Line 137).

The sentence on Page 8 Line 162 has been modified as follows: "*The control experiment (CTRL) has the original BB emissions, while the sensitivity experiment (BBER) has the March BB emissions reduced to 15% (Fig. 2b).*"

Figure R3 shows the global distribution of annual averaged BB emission and its monthly variation from four important biomass burning regions (Ding et al., 2021). As the reviewer speculated, the total BB emission in the ICP in March–April is only about 20% of that in South Africa in June–August (Table R1). However, the cloud cover enhancement induced by the BB aerosols is similar (over 30%) in both regions, suggesting a much stronger aerosol effect on climate in the ICP (Ding et al., 2021).

[Figure]

**Figure R3: Biomass burning (BB) carbon emission and its seasonal patterns in main BB regions in the world. (a) Global distribution of annual averaged carbon emissions from BB. (b) Monthly variation of carbon emissions from four important BB regions: Southeast Asia, North Africa, South Africa, and the Amazon.   After Ding et al. (2021).**

**Table R1. Averaged carbon emission in four regions with intensive BB activities during 2000-2015.   After Ding et al. (2021).**

| Zone | Period | Total Carbon Emission (Tg C) |
|---|---|---|
| Zone1 –Southeast Asia | Mar–Apr. | 62 |
| Zone2 –Center Africa | Nov–Dec. | 184 |
| Zone3 –South Africa | Jul–Aug. | 314 |
| Zone4 –South America | Aug–Sep. | 169 |

To provide a clearer picture of BB emission in the ICP region, we have added the sentence on Page 2 Line 57 as follows: "*Although the total BB emission in the ICP in March–April is only 20% of that in South Africa in June–August, the cloud cover enhancement induced by the BB aerosols is similar (over 30%) in both regions, suggesting a much stronger aerosol effect on climate in the ICP (Ding et al., 2021).*"

Although authors mentioned uncertainty in their modeling investigation, no uncertainty analysis was done. How about uncertainty in BB emissions?

**Responses:** To largely reduce the randomness and uncertainties introduced by initial and boundary conditions, we used ensemble simulations with six members by perturbing initial and boundary conditions and analysed the ensemble-mean results in this study. As concerned by the reviewer, we further analysed uncertainties of temperature, BC mass concentration and atmospheric heating rates (as shown in Figure 9a, Figure 12c and Figure 14c, respectively), and equivalent potential temperature, convective stability, and differences in convective stability (shown in Figure 13) in the revised manuscript.

As for the BB emissions, the uncertainty in emission inventories is mainly from a variety of measurements or analysis procedures, including detection of fire or areas burned, retrieval of fire radiative power, emission factors, biome types, burning stages, and fuel consumption estimates (Pan et al., 2020). The choice of emission inventory may markedly affect the simulated aerosols. For example, the magnitude of mean modelled smoke $PM_{2.5}$ can differ across five main inventories by >20 μg m$^{-3}$ in Singapore during the BB season (Liu et al., 2020b). The Quick Fire Emissions Dataset (QFED) provides a relatively large quantity of particle emissions from fires compared to the Fire INventory from NCAR (FINN) (Pan et al., 2020). While the comparison of BB emission inventories is beyond the scope of this study, the potential impact of using different inventories needs to be kept in mind. Note that the FINN version 1.5 utilized in this study is widely used in BB aerosol modelling investigations (Lee and Wang, 2020; Liu et al., 2020a; Wang et al., 2021; Takeishi and Wang, 2022). Also, our results are in agreement with many studies using other BB emission inventories. For example, the response of low cloud over subtropical Southeast Asia to March BB aerosols in this study is consistent with that of Ding et al. (2021), which used the QFED; the radiative response to BB aerosols over the ICP in this study is similar to that of Dong et al. (2019),

which used the Global Fire Emission Database v4.1; the BB aerosol-induced atmospheric circulation anomalies over the ICP agree with the results of Yang et al. (2022), which used the BB emission from the MERRA-2. It follows that although the choice of BB emission inventory may have an impact on the simulated results, it may not be sufficient to affect our qualitative conclusions.

As for the effects of BB emission reduction rate in the sensitivity experiment on our results, we have discussed them in response to the first comment. Briefly, the magnitude of the reduction rate does not influence our qualitative results, and more precise quantitative results need to be obtained by running more experiments in the future.

To consider the possible impact of BB emissions, we have added the following paragraph on Page-8 Line-152: "*Note that the choice of BB emission inventory could significantly affect the simulated aerosols due to the uncertainty in emission inventories introduced by a variety of measurements or analysis procedures, including detection of fire or areas burned, retrieval of fire radiative power, emission factors, biome types, burning stages, and fuel consumption estimates (Liu et al., 2020b; Pan et al., 2020). While the comparison of BB emission inventories is beyond the scope of this study, the FINN version 1.5 utilized in this study is widely used in BB aerosol modelling investigations (Lee and Wang, 2020; Liu et al., 2020a; Wang et al., 2021; Takeishi and Wang, 2022); nevertheless, the potential impact of using different inventories needs to be kept in mind.*"

**References**

Chavan, P., Fadnavis, S., Chakroborty, T., Sioris, C. E., Griessbach, S., and Müller, R.: The outflow of Asian biomass burning carbonaceous aerosol into the upper troposphere and lower stratosphere in spring: radiative effects seen in a global model, Atmos. Chem. Phys., 21, 14371-14384, 10.5194/acp-21-14371-2021, 2021.

Ding, K., Huang, X., Ding, A., Wang, M., Su, H., Kerminen, V.-M., Petäjä, T., Tan, Z., Wang, Z., Zhou, D., Sun, J., Liao, H., Wang, H., Carslaw, K., Wood, R., Zuidema, P., Rosenfeld, D., Kulmala, M., Fu, C., Pöschl, U., Cheng, Y., and Andreae, M. O.: Aerosol-boundary-layer-monsoon interactions amplify semi-direct effect of biomass smoke on low cloud formation in Southeast Asia, Nat. Commun., 12, 6416, 10.1038/s41467-021-26728-4, 2021.

Dong, X., Fu, J. S., Huang, K., Zhu, Q., and Tipton, M.: Regional Climate Effects of Biomass Burning and Dust in East Asia: Evidence From Modeling and Observation, Geophys. Res. Lett., 46, 11490-11499, 10.1029/2019gl083894, 2019.

Lee, D., Sud, Y. C., Oreopoulos, L., Kim, K. M., Lau, W. K., and Kang, I. S.: Modeling the influences of aerosols on pre-monsoon circulation and rainfall over Southeast Asia, Atmos. Chem. Phys., 14, 6853-6866, 10.5194/acp-14-6853-2014, 2014.

Lee, H. H., and Wang, C.: The impacts of biomass burning activities on convective systems over the Maritime Continent, Atmos. Chem. Phys., 20, 2533-2548, 10.5194/acp-20-2533-2020, 2020.

Liu, L., Cheng, Y., Wang, S., Wei, C., Pöhlker, M. L., Pöhlker, C., Artaxo, P., Shrivastava, M., Andreae, M. O., Pöschl, U., and Su, H.: Impact of biomass burning aerosols on radiation, clouds, and precipitation over the Amazon: relative importance of aerosol–cloud and aerosol–radiation interactions, Atmos. Chem. Phys., 20, 13283-13301, 10.5194/acp-20-13283-2020, 2020a.

Liu, T., Mickley, L. J., Marlier, M. E., DeFries, R. S., Khan, M. F., Latif, M. T., and Karambelas, A.: Diagnosing spatial biases and uncertainties in global fire emissions inventories: Indonesia as regional case study, Remote Sens. Environ., 237, 111557, https://doi.org/10.1016/j.rse.2019.111557, 2020b.

Pan, X., Ichoku, C., Chin, M., Bian, H., Darmenov, A., Colarco, P., Ellison, L., Kucsera, T., da Silva, A., Wang, J., Oda, T., and Cui, G.: Six global biomass burning emission datasets: intercomparison and application in one global aerosol model, Atmos. Chem. Phys., 20, 969-994, 10.5194/acp-20-969-2020, 2020.

Takeishi, A., and Wang, C.: Radiative and microphysical responses of clouds to an anomalous increase in fire particles over the Maritime Continent in 2015, Atmos. Chem. Phys., 22, 4129-4147, 10.5194/acp-22-4129-2022, 2022.

Thornhill, G. D., Ryder, C. L., Highwood, E. J., Shaffrey, L. C., and Johnson, B. T.: The effect of South American biomass burning aerosol emissions on the regional climate, Atmos. Chem. Phys., 18, 5321-5342, 10.5194/acp-18-5321-2018, 2018.

Wang, J., jiang, Q., You, Y., Rao, X., Sheng, L., Gui, H., Hua, C., and Zhang, B.: Effects of Biomass Burning Aerosol in Southeast Asia on Haze and Precipitation over China, Meteor. Mon. (in Chinese), 47, 348-358, 2021.

Yang, S., Lau, W. K. M., Ji, Z., Dong, W., and Yang, S.: Impacts of radiative effect of pre-monsoon biomass burning aerosols on atmospheric circulation and rainfall over Southeast Asia and southern China, Clim. Dynam., 10.1007/s00382-021-06135-7, 2022.

---

## Author Comment (AC2)

**Responses to Anonymous Referee #2**

This paper uses WRF-Chem model to explore the impact of biomass burning (BB) on atmospheric circulation and precipitation during the peak biomass burning season (March) in the Indochina Peninsula (ICP). Authors utilize observations to show that March BB aerosols can reduce precipitation over the ICP in March but increase precipitation from April 1-20, indicating the long-lasting effects of March BB aerosols on precipitation, but with opposite effects in the two months. However, it is hard to determine the causality between BB aerosols over the ICP and atmospheric circulation (and precipitation), just from observations. Therefore, two groups of WRF-Chem experiments: with control (CTRL) and sensitivity (BBER) model scenarios, were performed to discern the mechanisms responsible for these feedbacks of BB aerosols on precipitation. To discern the feedback effects.

The paper is very well-written with brevity and high-quality visualization of all results. The manuscript should be published once the following comments are addressed:

Responses: We greatly appreciate these comments and suggestions. The manuscript has been improved by considering these comments and suggestions. Our detail responses are given point by point below in blue. The revised text is highlighted in red.

**Specific comments:**

**1) Section 4.1: Evaluation of model results**

For comparison of CTRL AOD and Precipitation with MODIS and TRMM observations respectively, if possible, please provide some domain-wide statistical difference metrics (such as mean bias, mean error, RMSE, correlation, etc.). This will be helpful to quantify the predictive capability of the default model for the ICP region in this study period.

**Responses**: As suggested by the reviewer, we have evaluated statistics metrics (mean observation, mean simulation, mean bias, relative mean bias, RMSE, and pattern correlation) for AOD, precipitation and 850-hPa wind. We present them in Table S1 below (also in the supplementary):

| Variable                                             | Mean Obs. | Mean
Sim. | Mean
Bias | Relative
Mean Bias
(%) | RMSE | Pattern
Correlation |
|------------------------------------------------------|-----------|--------------|--------------|------------------------------|------|------------------------|
| AOD                                                  | 0.44      | 0.33         | -0.11        | -25.63                       | 0.23 | 0.71                   |
| Precipitation (mm day -1 )                | 2.22      | 3.52         | 1.29         | 58.25                        | 2.67 | 0.71                   |
| 850-hPa zonal wind (m s <math>-1</math> ) | -0.17     | -0.55        | -0.38        | 230.5                        | 1.8  | 0.94                   |
| 850-hPa
meridional
wind (m s -1 )   | 0.44      | 0.30         | 0.14         | -30.85                       | 1.53 | 0.67                   |

Table S1. Evaluation Statistics for AOD, precipitation and 850-hPa wind.

To present the model evaluation more quantitatively, we revised the sentence on Page 11 Line 236 as follows: "*The WRF-Chem ensemble-mean rainfall based on six CTRL members (Fig. 4b) shows a spatial pattern consistent with that in the TRMM and the pattern correlation is up to 0.71, although the model overestimates the convection in the northern tropical Indian Ocean, orographic precipitation in the northwestern ICP region, and rainfall south of Japan*".

We have also added this sentence on Page 11 Line 238 in our revised manuscript: "*It* was reported that regional climate models, including the WRF, tend to overestimate precipitation due to deficiencies within the convective cloud and microphysical schemes (Caldwell et al., 2009; Argüeso et al., 2012)".

The sentence on Page 12 Line 243 has been revised as follows: "In general, the model can reasonably capture these observed circulation features with the pattern correlation of 0.94 and 0.67 for 850-hPa zonal and meridional wind components, respectively".

The sentence on Page 12 Line 250 has been modified as follows: "*The spatial pattern* of modeled AOD is consistent with MODIS satellite retrieval, with the pattern correlation of 0.71."

The following sentence has been added on Page 12 Line 252: "*The model simulation underestimates the AOD by 25.63% for the whole domain.*"

We have revised the last paragraph of this section on Page 13 Line 259 as follows:

"Generally, the model reproduces well the spatial distributions of rainfall, circulation and aerosols. Specific evaluation statistics are summarized in Table S1. Given this, the ensemble-mean differences between CTRL and BBER (i.e. CTRL minus BBER) are used to examine the effects of BB aerosols and associated physical mechanisms."

2) Besides, discerning the direct and indirect effects via sensitivity experiments with and without direct or indirect effects, doing an HYPLIT trajectory analysis to look at BB emissions trajectories in the modeling domain for the March-April period, if possible is encouraged. Trajectories of air mass relative to the black box that outlines the main Indochina Peninsula (ICP; 93°–110°E, 10°–24°N), may aid the authors' current inferences more in explaining the opposite impacts in March vs April.

**Responses:** We have performed the HYPLIT model 96h (4 days) air mass trajectories forward run using the 1-degree GDAS meteorological data at four high BB emission sites (95.75°E, 18.75°N; 102°E, 18.25°N; 106.5°E, 15°N; 105.5°E, 12.5°N) representing northwestern Indochina Peninsula, northern Indochina Peninsula, mid-eastern Indochina Peninsula and southern Indochina Peninsula, respectively. The multiple iterations of the trajectory were calculated from March 1st to April 20th 2010 at every six hours. The cluster mean trajectories of air mass and their frequency analysis are showed in supplementary (Fig. S1). It is worth noting that we also only did the trajectory analysis for March, since the results were quite similar to those for March 1st–April 20th.

To better explain the cause of AOD anomaly pattern in Figure 5a, we have added the following text on Page 13 Line 265: "The aerosol loading anomaly gradually decreased from northern ICP through the northern SCS up to the Northwest Pacific and the anomaly also declined westward from the ICP to the central Bay of Bengal (Fig. 5a). These are the results of BB aerosol dispersion downstream along with the subtropical westerlies and tropical easterlies. Lagrangian dispersion modelling for air mass shows that aerosols over the northern ICP can be transported to the northern SCS and southern China, while the aerosols over the southern ICP have westward trajectories of 11%–31% and partially reach the central Bay of Bengal (Fig. S1)"